



# Ozone-vegetation feedback through dry deposition and isoprene emissions in a global chemistry-carbon-climate model

Cheng Gong[1, 2], Yadong Lei[2, 3], Yimian Ma[2, 3], Xu Yue[4*] and Hong Liao[4*]

[1]State Key Laboratory of Atmospheric Boundary Layer Physics and Atmospheric Chemistry (LAPC), Institute of

Atmospheric Physics, Chinese Academy of Sciences, Beijing, 100029, China

[2]University of Chinese Academy of Sciences, Beijing, 100029, China

[3]Climate Change Research Center, Institute of Atmospheric Physics, Chinese Academy of Sciences, Beijing 100029, China

[4]Jiangsu Key Laboratory of Atmospheric Environment Monitoring and Pollution Control, Jiangsu Collaborative Innovation

Center of Atmospheric Environment and Equipment Technology, School of Environmental Science and Engineering,

Nanjing University of Information Science and Technology, Nanjing, 210044, China

*Correspondence to*: Xu Yue (yuexu@nuist.edu.cn) and Hong Liao (hongliao@nuist.edu.cn)

**Abstract.** Ozone-vegetation feedback is essential to tropospheric ozone ($O_3$) concentrations. The $O_3$ stomatal uptake damages leaf photosynthesis and stomatal conductance and, in turn, influences $O_3$ dry deposition. Further, $O_3$ directly influences isoprene emissions, an important precursor of $O_3$. The effects of $O_3$ on vegetation further alter local

meteorological fields and indirectly influence $O_3$ concentrations. In this study, we apply a fully coupled chemistry-carbon-climate global model (ModelE2-YIBs) to evaluate changes in $O_3$ concentrations caused by $O_3$–vegetation interactions. Different parameterizations and sensitivities of the effect of $O_3$ damage on photosynthesis, stomatal conductance, and isoprene emissions (IPE) are implemented in the model. The results show that $O_3$-induced inhibition of stomatal conductance increases surface $O_3$ on average by +2.1 (+1.4) ppbv in eastern China, +1.6 (-0.5) ppbv in the eastern U.S., and +1.3 (+1.0)

ppbv in western Europe at high (low) damage sensitivity. Such positive feedback is dominated by reduced $O_3$ dry deposition, in addition to the increased temperature and decreased relative humidity from weakened transpiration. Including the effect of $O_3$ damage on IPE slightly reduces surface $O_3$ concentrations by influencing precursors. However, the reduced IPE weakens surface shortwave radiative forcing of secondary organic aerosols leading to increased temperature and $O_3$ concentrations in the eastern U.S. This study highlights the importance of interactions between $O_3$ and vegetation with regard to $O_3$

concentrations and the resultant air quality.

## 1 Introduction

Tropospheric ozone ($O_3$) is generated by photochemical reactions involving nitrogen oxides ($NO_x$) and volatile organic compounds (VOCs) under strong solar radiation (Sillman, 1999; Atkinson, 2000; Jacob and Winner, 2009). It is one of the most important air pollutants and has been of widespread concern (Wang et al., 2017; Li et al., 2019). High $O_3$

concentrations at the surface can not only injure human respiratory health (Gauderman et al., 2004; Lelieveld et al., 2015)





but also lead to considerable damage to plants and crops, which further changes the land carbon budget (Fuhrer et al., 1997;Yue and Unger, 2014; Lombardozzi et al., 2015). In turn, vegetation can modulate $O_3$ concentrations via influencing dry deposition processes, precursor emissions (such as those of isoprene, monoterpene and sesquiterpene) and meteorological fields. Studying $O_3$–vegetation interactions is of great importance to better understand the variations in $O_3$

concentrations as well as the ecosystem carbon cycle, particularly for regions with high $O_3$ levels and vegetative cover.

Ground-level $O_3$ reduces vegetation photosynthesis by stomatal uptake (Fuhrer et al., 1997; Ainsworth et al., 2012). Through a globally statistical meta-analysis, Wittig et al. (2007) showed that the elevated $O_3$ since the preindustrial period depressed photosynthesis and stomatal conductance of trees by 9-13% and 11-15%, respectively. A recent global meta-analysis on

poplar showed that current $O_3$ concentrations reduced the $CO_2$ assimilation rate and stomatal conductance by 33% and 25%, respectively, compared to that of charcoal-filtered air (Feng et al., 2019a). In model studies, an off-line process-based vegetation model (the Yale Interactive Terrestrial Biosphere model, or YIBs) estimated that present-day effect of $O_3$ damage reduced gross primary productivity (GPP) by 4-8% on average over the eastern US during the summer (Yue and Unger, 2014) and annual net primary productivity (NPP) by approximately 14% in China (Yue et al., 2017). Lombardozzi et al.

(2015) also showed that the present-day $O_3$ exposure reduces GPP globally by 8–12% using the Community Land Model (CLM).

Isoprene emissions (IPE) from vegetation can be affected by surface $O_3$. Isoprene is the most dominant species among biogenic VOCs (BVOCs) and accounts for approximately one-half of global BVOC emissions (Guenther et al., 2012). The

effect of $O_3$ on IPE is complex. Calfapietra et al. (2009) reviewed observational experiments in Italy and proposed a hypothesis that there might be a detoxification effect resulting from $O_3$–IPE interactions. Vegetation under a low accumulated $O_3$ dose can be simulated to increase the levels of IPE to reduce oxidative damage, but months of $O_3$ exposure are harmful to metabolism and reduce IPE. Several studies have showed that $O_3$ fumigation over a short time (days to weeks) but at high concentrations (100-300 ppbv) led to increased IPE (Velikova et al., 2005; Fares et al., 2010), while some other

experiments conducted over an entire growing season (at least 3 months) under controlled $O_3$ concentrations (approximately 80 ppbv) showed that $O_3$ reduced IPE (Calfapietra et al., 2008; Yuan et al., 2016; Yuan et al., 2017). A recent global meta-analytic review showed that IPE negatively responded to elevated $O_3$ (91 ppbv on average) by -8% (Feng et al., 2019b). Overall, consecutive exposure to high $O_3$ levels has a negative impact on IPE, although there are large uncertainties resulting from vegetation type (Tiiva et al., 2007; Ryan et al., 2009), temperature (Hartikainen et al., 2009) and $CO_2$ concentration

(Calfapietra et al., 2008).

Vegetation can affect $O_3$ concentrations through stomatal uptake (the majority of $O_3$ dry deposition). Val Martin et al. (2014) showed that the $O_3$ dry deposition velocity in the Community Earth System Model (CESM) significantly increased and was more reasonable when the original scheme (Wesely, 1989), which assumed that stomatal resistance was only related to



temperature and water vapor, was replaced with a scheme coupled to vegetation (Collatz et al., 1991; Sellers et al., 1996). In addition, BVOC emissions can change the local $NO_x$/VOC ratio and, in turn, influence $O_3$ concentrations. For example, Fu and Liao (2012) showed that the interannual variations in BVOCs alone can lead to 2-5% differences in simulated $O_3$ over China during the summer using the Model of Emissions of Gases and Aerosols from Nature (MEGAN) (Guenther et al.,

2006) module embedded within the global three-dimensional chemical transport model (GEOS-Chem). Calfapietra et al. (2013) reviewed the role of BVOCs emitted by urban trees on $O_3$ concentrations in cities and showed that BVOCs generally promoted $O_3$ formation because of the VOC-limited condition. Furthermore, the modifications of meteorological fields caused by vegetation (Liu et al., 2006; Wu et al., 2011) may also potentially have an effect on $O_3$ formation as well as vegetation growth. As a result, $O_3$ stomatal uptake ($O_3$ dry deposition via stomata), BVOC emissions and changes in

meteorological fields are connected and jointly affect $O_3$ concentrations.

Thus far, very few studies have comprehensively investigated the $O_3$-vegetation feedback on a global scale. Sadiq et al. (2017) investigated the effect of $O_3$ damage on the photosynthesis rate and stomatal conductance as well as potential meteorological feedback on surface $O_3$ concentrations using the CESM. They found that $O_3$–vegetation interactions led to

increased $O_3$ concentrations mainly in Europe, the northern U.S. and North China. However, the effect of $O_3$ on BVOCs was not directly considered but was indirectly simulated by the increased temperature resulting from $O_3$–vegetation interactions. The $O_3$ damage sensitivities for photosynthesis and stomatal conductance were calculated by using two decoupled linear regressions with accumulated $O_3$ concentrations. However, the linear slope of the photosynthetic rate and stomatal conductance to $O_3$ was zero for some vegetation types (such as broadleaf forests), showing significant effect of $O_3$ damage

even at zero $O_3$ concentrations. Based on the same flawed $O_3$ damage scheme, Zhou et al. (2018) calculated responses of leaf area index (LAI) to surface $O_3$ and implemented steady-state results for the GEOS-Chem model to simulate $O_3$ perturbations. Such asynchronous coupling may underestimate $O_3$ changes caused by the full pollution–biosphere interactions, not to mention the omission of feedback of $O_3$ to BVOC emissions and meteorology. More comprehensive work utilizing a validated $O_3$ damage scheme and considering the direct effect of $O_3$ on BVOCs is necessary to reasonably predict $O_3$-

vegetation feedback on $O_3$ concentrations.

In this study, we apply a semimechanistic $O_3$ damage scheme (Sitch et al., 2007) to the YIBs dynamic vegetation model coupled with the global Earth system model NASA ModelE2 (ModelE2-YIBs) to explore $O_3$-induced changes in stomatal conductance and evaluate the consequences of such changes on surface $O_3$ concentrations ($O_3$-vegetation feedback via $O_3$

dry deposition). Then, two schemes are proposed to estimate $O_3$ damage to IPE based on the existing scheme generated by Sitch et al. (2007) and that from observations. The feedback of $O_3$ damage to both stomatal conductance and IPE and the resultant effect on surface $O_3$ concentrations is calculated by using ModelE2-YIBs. Finally, the associated meteorological feedback to $O_3$ concentrations is discussed. We found that the $O_3$-vegetation feedback enhanced surface $O_3$ concentrations particularly in $O_3$-polluted regions.



## 2 Method

### 2.1 The NASA ModelE2-YIBs model

NASA ModelE2-YIBs is a fully coupled chemistry-carbon-climate global model with a horizontal resolution of 2° latitude × 2.5° longitude and 40 vertical layers up to 0.1 hPa. The dynamic and physical processes are calculated every 30 minutes.

Gas-phase chemistry in the troposphere includes basic $NO_x$-$HO_x$-$O_x$-CO-$CH_4$ chemistry as well as peroxyacyl nitrates and the following hydrocarbons: terpenes, isoprene, alkyl nitrates, aldehydes, alkenes, and paraffins. Chlorine-containing and bromine-containing compounds, chlorofluorocarbons (CFC) and $N_2O$ source gases are all included in the stratospheric gas-phase chemistry. Dry deposition of gases is calculated by using a resistance-in-series scheme, which was updated to include coupling to stomatal resistance (Val Martin et al., 2014). In addition, the model interactively simulates aerosols such as

sulfate, nitrate, elemental and organic carbon, sea salt and dust considering the climate through direct (Koch et al., 2006) and indirect effects (Menon et al., 2008; Menon et al., 2010) and gas-phase chemistry by affecting photolysis rates (Bian et al., 2003). Meteorological and hydrological variables in this model have been fully validated via observations and a reanalysis dataset (Schmidt et al., 2014). The anthropogenic emission inventory for the present-day (2010) from the IPCC RCP8.5 scenario (van Vuuren et al., 2011) is utilized in this study.

The YIBs model is a dynamic vegetation model that includes 9 plant functional types (PFTs) (Table S1) and can simulate biophysical processes of photosynthesis, transpiration and respiration with variations in meteorological fields. Leaf photosynthesis and stomatal conductance are calculated by using the Farquhar and Ball–Berry models (Farquhar et al., 1980) as follows:

$$A_{tot} = \min(J_c, J_e, J_s) \tag{1}$$

$$g_s = m \frac{(A_{tot} - R_d) \times RH}{c_s} + b \tag{2}$$

where the total leaf photosynthesis ($A_{tot}$) is the minimum value of the ribulose-1,5-bisphosphate carboxylase (RuBisCO)-limited rate of carboxylation ($J_c$), light-limited rate ($J_e$), and export-limited rate ($J_s$). Stomatal conductance ($g_s$) is calculated by the $A_{tot}$, dark respiration rate ($R_d$), relative humidity ($RH$) and $CO_2$ concentration at the leaf surface ($c_s$). The values of $m$

and $b$ are different for different PFTs (Table S1). A canopy radiation scheme is applied in YIBs to separate diffuse and direct light for sunlit and shaded leaves. The LAI and tree growth are dynamically simulated with the allocation of carbon assimilation. Carbon fluxes, phenology, LAI, GPP, and net ecosystem exchange (NEE), as well as other parameters of vegetation in ModelE2-YIBs, have been previously extensively evaluated and agree well with the observations (Yue and Unger, 2015).


The $O_3$ dry deposition velocity ($V_d$) in ModelE2-YIBs are calculated following the multiple-resistance approach originally described by Wesely (1989):





$$V_d = \frac{1}{R_a + R_b + R_c} \tag{3}$$

where $R_a$, $R_b$ and $R_c$ are the aerodynamic resistance, quasi-laminar sublayer resistance above canopy, and surface resistance, respectively. $R_c$ is computed as follows:

$$\frac{1}{R_c} = \frac{1}{R_s} + \frac{1}{R_{lu}} + \frac{1}{R_{cl}} + \frac{1}{R_g} \tag{4}$$

where $R_s$, $R_{lu}$, $R_{cl}$ and $R_g$ represent the stomatal resistance, leaf cuticle resistance, lower canopy resistance and the ground resistance, respectively. In this study, the original parameterization for $R_s$, which is empirically expressed by solar radiation, surface air temperature, and the molecular diffusivities for water vapor, has been substituted by the reciprocal of $g_s$ from Eq. (2) following Val Martin et al. (2014). In this case, $O_3$ dry deposition can be interactively influenced by the stomatal $O_3$ uptake process for vegetation.

Isoprene and α-pinene are considered as the precursors for biogenic secondary organic aerosols (SOA) in ModelE2-YIBs, which are online computed based on the two-product scheme developed by Chung and Seinfeld (2002) . The mass-based yields of semivolatile species are set following Presto et al. (2005). Temperature ($T$) dependence on partitioning coefficient ($K_p$) are given by the Clausiuse-Clapeyron equation:

$$K_p = K_{sc} \frac{T}{T_{sc}} exp \left[ \frac{\Delta H}{R} \left( \frac{1}{T} - \frac{1}{T_{sc}} \right) \right] \tag{5}$$

where ΔH is the enthalpy of vaporization and is set as 42.0 kJ mol$^{-1}$ for isoprene (Chung and Seinfeld, 2002; Henze and Seinfeld, 2006) and 72.9 kJ mol$^{-1}$ for α-pinene. $K_{sc}$ is the saturation concentrations at the temperature $T_{sc}$ (295 K) and set as 1.62 (0.064) m$^3$ μg$^{-1}$ and 0.0086 (0.0026) m$^3$ μg$^{-1}$ for the two products formed by oxidation of isoprene (α-pinene), respectively (Presto et al., 2005; Henze and Seinfeld, 2006).

**2.2 Schemes of $O_3$ damage to vegetation**

**2.2.1 The effect of $O_3$ damage to photosynthesis and stomatal conductance**

A semi-mechanistic scheme proposed by Sitch et al. (2007) is applied in this study that simulates the effect of $O_3$ damage to the photosynthesis rate via the following formula:

$$A_{totd} = F \times A_{tot} \tag{6}$$

where $A_{totd}$ and $A_{tot}$ are the $O_3$-damaged and original total leaf photosynthesis, respectively, and $F$ is an "uptake of $O_3$ factor" depending on the instantaneous leaf uptake of $O_3$ as follows:

$$F = 1 - a \times \max \left[ F_{O_3} - F_{O_3, crit}, 0.0 \right] \tag{7}$$

where $F_{O3}$ is the $O_3$ uptake rate by the stomata and $F_{O3,crit}$ is the critical vegetation-dependent threshold for damage. Parameter $a$ has two series of values for different vegetation types representing high or low sensitivities to $O_3$ (Table S1).

Moreover, damaged photosynthesis leads to a lower stomatal conductance via formula (2) to aid vegetation resistance to potential $O_3$ damage. This scheme has been utilized in many previous studies, which have reported that $O_3$ reduces GPP by





4–8% on an annual mean basis in the eastern U.S. and by 10-20% during the summer in China (Yue and Unger, 2014; Yue et al., 2017).

### 2.2.2 The effect of O₃ damage to IPE

A photosynthesis-based scheme is applied in the model to simulate IPE, which is the function of the $J_e$, canopy temperature and intercellular $CO_2$ concentration (Arneth et al., 2007; Unger, 2013). To date, there are no mature parameterizations that calculate the contributions of O₃ damage to IPE. Here, we propose two schemes based on observations to quantify the changes in surface O₃ concentrations resulting from O₃ damage to IPE.

The first scheme assumes that O₃ leads to the same percentage of damage to photosynthesis and IPE because IPE are observed to linearly vary with photosynthesis (Yuan et al., 2016). The affected IPE ($IPE_d$) can be calculated as follows:

$$IPE_d = F \times IPE \tag{8}$$

where F is calculated by using Eq. (7) and *IPE* is the original level of IPE. Hereafter, this scheme is termed the "F scheme."

Another scheme is based on open-top chamber (OTC) observations. Although many experiments have studied the effects of O₃ on IPE, most have applied a limited range of O₃ levels (e.g., 7.3-56.6 ppbv in Hartikainen et al. (2009) or >100 ppbv in Fares et al., (2010)). In reality, surface O₃ concentrations can vary from several parts-per-billion-volume (e.g., in the polar region during the winter) to over 100 ppbv (e.g., in megacities of China during the summer). To date, only one study (Yuan et al., 2017) has explored the responses of IPE to different levels of O₃ damage for two poplar clones; a linear regression between the percentage damage of IPE (*PDI*) and the cumulative stomatal uptake of O₃ > 1 nmol O₃ m⁻² s⁻¹ (*POD₁*) was derived as follows:

$$PDI = (-0.0086 \times POD_1 + 1.0194) \times 100\% \tag{9}$$

The *POD₁* is calculated by the following formula:

$$POD_1 = \int_{i=1}^{n} (F_{O_3} - 1) dt \tag{10}$$

where $F_{O3}$ is the O₃ uptake rate by stomata (nmol O₃ m⁻² s⁻¹), which is the same as that in Eq. (7). In this study, the *POD₁* accumulated over the growth season, which is defined as April to October north of 23.5°N (e.g., Tucker et al., 2001; White et al., 2002; Yin et al., 2014; Wang et al., 2019), November to March south of 23.5°S (e.g., Broich et al., 2015; Moore et al., 2016), and 200 days between 23.5°N to 23.5°S because the leaf phenology in tropical evergreen forests is not determined by seasonality (Xiao et al., 2006). We apply the same PDI function (Eq. (9)) of poplar for all vegetation types because of the data limit as follows:

$$IPE_d = \min (PDI, 100\%) \times IPE \tag{11}$$

Hereafter, this scheme is termed a "linear scheme." Different from the F scheme, the linear scheme calculates IPE damage using accumulated O₃ instead of instantaneous O₃ concentrations.



### 2.3 Descriptions for sensitivity experiments

Seven experiments (Table 1) are conducted to explore the feedback of vegetation on surface $O_3$ concentrations via influencing $O_3$ dry deposition, IPE, as well as meteorological fields. The control simulation (CTRL) does not include the effect of $O_3$ damage to vegetation. Two cases (DRY_high and DRY_low) are established to investigate the feedback via $O_3$

dry deposition with high or low $O_3$ damage sensitivities ($a$ in Eq. (7)). Then, the effect of $O_3$ damage to IPE is added by using either F or linear schemes, resulting in four more experiments (TOTAL_F_high, TOTAL_F_low, TOTAL_LINEAR_high, and TOTAL_LINEAR_low). In the CTRL run, offline $O_3$ damage to photosynthesis and stomatal conductance are calculated using the Sitch et al. (2007) scheme; offline $O_3$ damage to IPE is calculated using the linear scheme. The offline $O_3$ damage to IPE produced by using the F scheme is calculated in DYR_high and DYR_low. For these

offline simulations, $O_3$-induced damage is not fed back to affect the climate, chemistry, or carbon cycle.

For each experiment, 20-year simulations are performed with 5 initial spin-up years. Outputs of the last 15 years are averaged and analyzed. Regionally, the results in the eastern U.S. (30°N-45°N, 75°W-90°W), western Europe (35°N-60°N, 10°W-20°E) and eastern China (20°N-40°N, 105°E-122°E) are compared and discussed.

### 2.4 Observed ground-level $O_3$ network and model evaluation

To evaluate simulated $O_3$ concentrations, three observational networks are utilized as follows: the Air Quality Monitoring Network from Ministry of Ecology and Environment (AQMN-MEE) in China, Clean Air Status and Trends Network (CASTNET) in the U.S. and European Monitoring and Evaluation Programme (EMEP) in Europe. The summer concentrations for CASTNET and EMEP are averaged over the year 2010 but those for AQMN-MEE are averaged over

2014 because this network was established in 2013 and started to provide high-quality data beginning in 2014. The simulated $O_3$ concentrations are interpolated in the observational sites by using a bilinear interpolation method. Normalized mean biases (NMBs) are calculated by using the following equation:

$$NMB = \sum_i^n (S_i - O_i) / \sum_i^n O_i * 100\%$$ (12)

where $S_i$ and $O_i$ are the simulated and observed $O_3$ concentrations, respectively, and $n$ is the total number of observational

sites.

### 3 Results

### 3.1 CTRL simulation and model evaluation

Figure 1 shows a comparison of the simulated summer $O_3$ concentrations to the observations. The model in general captures reasonable spatial patterns with a correlation coefficient of 0.41, although it overestimates $O_3$ concentrations by 29.3%. The

NMB between simulations and observations in U.S and Europe is 11.7% and 13.2%, respectively, which is comparable with





the simulation performed by CESM (Lamarque et al., 2012; Sadiq et al., 2017). The large overestimate is mainly a result of overestimation in China, where the anthropogenic emissions has been rapidly changing in recent years (Zheng et al., 2018). As a result, the surface $O_3$ concentrations may show large differences between the year of 2010 and 2014. Also, the rural $O_3$ concentrations are likely higher than urban $O_3$ concentrations because of the $NO_x$ titration effect in cities. However, most of

the observational sites in AQMN-MEE are located in urban area, which might be another reason for the surface $O_3$ overestimates in China (Yue et al., 2017).

To further compare the performance of ModelE2-YIBs with other chemistry-climate models, we select six simulated cases performed by different model members in Atmospheric Chemistry and Climate Model Intercomparision Project (ACCMIP) (Lamarque et al., 2013) and implement the evaluation with the same observational data (Fig. S1). The correlation coefficient

(0.41) and NMB (29.3%) for ModelE2-YIBs are within the ranges of 0.36 to 0.60 and -16.0% to 45.1% by the model ensembles, suggesting that ModelE2-YIBs has comparable performance with other state-of-the-art models. However, most of the current chemistry-climate models lack the interactive coupling between vegetation and chemistry, which is the main focus of this study. The vegetation variables (e.g. GPP and LAI) in ModelE2-YIBs have been fully evaluated in previous studies (Yue and Unger, 2015), making ModelE2-YIBs a suitable tool for this work.

Figure 2 shows the global June-July-August (JJA) surface $O_3$ concentrations, $O_3$ dry deposition velocity, GPP and IPE. Simulated $O_3$ is high in the eastern U.S., western Europe, India, and eastern China (Fig. 2a). The spatial pattern of $O_3$ dry deposition velocity (Fig. 2b) resembles that of the GPP (Fig. 2c) because the $O_3$ stomatal uptake dominantly contributed to the dry deposition. Both are high in the eastern U.S., western Europe, Amazon, eastern China, and Indonesia and show a

reasonable magnitude consistent with previous modeling studies (Val Martin et al., 2014; Yue and Unger, 2015; Sadiq et al., 2017). The spatial pattern of IPE (Fig. 2d) also resembles that of the GPP (Fig. 2c), except that the IPE in Europe are lower than those in other regions. Such discrepancies are likely attributed to the lower fraction of deciduous broadleaf forest, which provides a high yield of IPE (Potter et al., 2001).

**3.2 Offline $O_3$ damage to IPE**

Figure 3 shows the effect of $O_3$ damage to IPE during the boreal summer. For different schemes, reductions in IPE show a similar spatial distribution with significant damages in the eastern U.S., western Europe, and eastern China, where both $O_3$ concentrations and vegetative cover are high. For the F scheme with high sensitivity, the damage mediated by the IPE can reach as high as 30% in eastern China and > 20% in the eastern U.S. and western Europe (Fig. 3a). However, the F scheme with low sensitivity predicts a low damage of ~10% in these regions (Fig. 3b). On a global scale, IPE decreases by 1.2-3.2%

because of the $O_3$ effect. The damage using the linear scheme is generally within the low-to-high range of predictions by using the F schemes. For the linear scheme, IPE in eastern China show the greatest damage of ~15%.





Figure 4 shows seasonal variations in the effect of $O_3$ damage to IPE in eastern China, the eastern U.S. and western Europe. The magnitude of IPE changes is generally within the range of 10-29%, as summarized by the observational meta-analysis (Feng et al., 2019b). The F scheme is dependent on instantaneous $O_3$ uptake, which peaks during the summer when both

surface $O_3$ and stomatal conductance are high. In contrast, the linear scheme depends on the accumulated $O_3$ flux, which is increases from zero to high levels during the growth season. As shown, the percentage of $O_3$ damage to IPE is low during April and May but increases to a similar magnitude as that in the F scheme with high sensitivity during August; it reaches a maximum in October. The differences in the F (instantaneous) and linear (accumulated) schemes cause distinct seasonal variations in the IPE damage, which might cause different feedback to the $O_3$ concentrations. However, the IPE peaks during

summer (Fig. S2), suggesting that absolute changes in IPE are most significant during this season. As a result, we focus our analyses on the summer to explore the $O_3$-vegetation interactions and feedback.

### 3.3 $O_3$-vegetation feedbacks on surface $O_3$ concentrations

The effect of $O_3$ damage to stomatal conductance inhibits dry deposition (Fig. S3) leading to significant increases in summer surface $O_3$, particularly in eastern China, Japan, the eastern U.S., and western Europe (Figs. 5a-b). The positive feedback can

be greater than 5 ppbv in eastern China with high sensitivity (Fig. 5a). Smaller changes are predicted for low sensitivity, which shows limited perturbations in the U.S. and Japan (Fig. 5b). Including the effect of $O_3$ damage to both stomatal conductance and IPE maintains the spatial pattern of $O_3$ changes but occurs at a lower magnitude (Figs. 5c-f) because these two effects offset each other. With high damage to stomatal conductance, surface $O_3$ remains increasing in eastern China, Japan, the eastern U.S., and western Europe even with reduced IPE (Figs. 5c and 5e). However, with low damage to stomatal

conductance, surface $O_3$ shows limited changes in Europe, China and Japan when IPE are simultaneously reduced (Figs. 5d and 5f). Surprisingly, surface $O_3$ increases over the eastern U.S. in these cases (Figs. 5d and 5f) compared to the limited changes when IPE remain unperturbed (Fig. 5b).

Figure 6 summarizes the changes in surface $O_3$ over sensitive regions. Without IPE feedback, the effect of $O_3$ damage to

stomatal conductance leads to changes in regionally averaged surface $O_3$ by +2.1 (+1.4) ppbv in eastern China, +1.6 (-0.5) ppbv in the eastern U.S., and +1.3 (+1.0) ppbv in western Europe for high (low) damage sensitivity. Changes in eastern China are the greatest compared to those of the other two regions, mainly because of the high $O_3$ level (Fig. 1a) and sensitive tree species (the high $a$ and low $F_{o3,crit}$ for deciduous broadleaf forest, Table S1). Surface $O_3$ is predicted to decrease in the eastern U.S. with the low damage sensitivity, though such a change is not significant over most grids (Fig. 5b). The inclusion

of the effect of $O_3$ damage for both stomatal conductance and IPE slightly weakens the $O_3$ feedback, leading to changes in $O_3$ concentrations of +1.7 (+0.4) ppbv with the F scheme and +2.1 (-0.1) ppbv with the linear scheme in eastern China for high (low) sensitivity. The regional maximum $O_3$ changes can reach 6.9 (3.9) ppbv in eastern China. Further, the effect of $O_3$ damage to IPE weakens the positive feedback in western Europe by approximately 1-2 ppbv. The negative $O_3$ changes in the




eastern U.S. with low $O_3$ damage are +0.9 (F scheme) or +0.8 (linear scheme) ppbv on average when IPE feedback is included.

Although damage to stomatal conductance and IPE exert opposite effects, surface $O_3$ in general increases after including

both processes (Fig. 6), suggesting that dry deposition inhibition plays the dominant role. For the same $O_3$ damage sensitivity to stomatal conductance, changes in surface $O_3$ remain similar over eastern China and the eastern U.S. between the F and linear schemes in terms of the responses of the IPE. However, responses in western Europe are weaker for the linear scheme (Fig. 5e) compared to that of the F scheme (Fig. 5c), though the former predicts lower reductions in IPE (Fig. 3). Nevertheless, inclusion of IPE reductions helps increase surface $O_3$ over the eastern U.S. (Figs. 5d/5f vs. Fig. 5b). These

changes cannot be explained by the changes in IPE but are indirectly related to $O_3$-vegetation feedback to meteorology.

### 3.4 Effects of $O_3$–vegetation interactions on meteorology and vegetation

Figure 7 and Figure 8 show the changes in surface air temperature and relative humidity (RH) between different sensitivity experiments and the CTRL simulation, respectively. When considering the effect of $O_3$ damage to stomatal conductance alone, eastern China becomes warmer (Fig.7a and 7b) and drier (Fig.8a and 8b), favoring $O_3$ chemical production and

increasing surface $O_3$ concentrations (Jacob and Winner, 2009). The damaged stomatal conductance weakens leaf-level transpiration and thus reduces the latent heat flux at the surface (Fig. S4), leading to a higher temperature and lower RH. The the effect of $O_3$ damages are weaker in the eastern U.S. and western Europe because of the lower $O_3$ concentrations, resulting in insignificant changes in temperature and RH over these regions.

The effect of $O_3$ damage to IPE has limited impacts on RH (as shown in Figs. 8c/e vs. 8a and Figs. 8d/f vs. 8b) but significantly increases surface air temperature in the eastern U.S. (as shown in Figs. 7c/e vs. 7a and Figs. 7d/f vs. 7b). The temperature in western Europe also slightly increases when IPE reductions are included, particularly when utilizing the F scheme with high sensitivity (Fig. 7c). Isoprene is among the most important precursors for the formation of SOAs (Claeys et al., 2004), which are able to reduce surface air temperature by light extinction (Charlson et al., 1992). As a result, the $O_3$-

induced reduction of IPE decreases SOA loading and weakens the "cooling effect" of aerosols, leading to a higher temperature at the surface. The positive changes in shortwave radiative forcing following SOA reduction are the strongest in the eastern U.S. when considering the effect of $O_3$ damage to IPE, particularly for the F schemes with high sensitivity (Fig. 9). Such warming explains why the reduced IPE helps increase the surface $O_3$ in the eastern U.S. (Fig. 6). However, aerosols in regions with high anthropogenic emissions (such as eastern China) are more dominated by inorganic components (Sun et

al., 2006; Yang et al., 2011); thus, the changes in SOAs are less important. As a result, the feedback of $O_3$-induced IPE reductions on temperature is not significant in eastern China compared to that of other regions.



In addition to the direct damage (Fig. 3), IPE are indirectly affected by perturbations in the LAI and meteorology. Figure S5 shows that the LAI decreases in three polluted regions (eastern China, the eastern U.S. and western Europe) because of the $O_3$-mediated inhibition of photosynthesis, although the magnitude is typically within 5%. Moderate changes in the LAI by $O_3$ have also been reported in previous studies (Yue and Unger, 2015; Sadiq et al., 2017), suggesting that LAI feedback is too low to effectively influence IPE and the consequent surface $O_3$. Furthermore, the warming effects resulting from the $O_3$-induced inhibition on stomatal conductance (Fig. 7) and the changes in the LAI cause limited changes in IPE (Fig. S6), suggesting that $O_3$-vegetation feedback does not significantly change IPE. In comparison, Sadiq et al. (2017) reported a strong positive feedback (3-5 times greater than our results) on IPE caused by increased temperature from reduced transpiration when the effect of $O_3$ damage to stomatal conductance is considered. However, Sadiq et al. (2017) might have overestimated temperature feedback because their parameterizations of $O_3$ damage to plants employ constant intercepts for some PFTs, which results in sustained damage even at low $O_3$ concentrations.

## 4 Conclusions and discussion

In this study, we explore the effect of $O_3$-vegetation feedback on surface $O_3$ concentrations by considering the effects of $O_3$ damage on photosynthesis, stomatal conductance, and IPE in a fully coupled global chemistry-carbon-climate model. Three regions with high $O_3$ levels and dense vegetation cover, including eastern China, the eastern U.S. and western Europe, are examined during the summer. The positive feedback increases $O_3$ concentrations on average by +2.1 (+1.4) ppbv in eastern China, +1.6 (-0.5) ppbv in the eastern U.S., and +1.3 (+1.0) ppbv in western Europe for high (low) $O_3$ damage to stomatal conductance and the consequent inhibition of dry deposition. Additionally, the effect of $O_3$ damage to stomatal conductance increases the surface temperature and decreases the RH by weakening transpiration, which favors $O_3$ chemical production and increases surface $O_3$ concentrations. Including the effect of $O_3$ damage to IPE slightly weakens the positive feedback in eastern China and western Europe but increases $O_3$ concentrations by 0.9-1.5 ppbv with the F scheme or 0.8-1.2 ppbv with the linear scheme in the eastern U.S. The main cause for this result is likely related to the increased temperature following reduced SOA concentrations. Our results show that $O_3$–vegetation interactions increase surface $O_3$ by reducing dry deposition (from inhibition of stomatal conductance) and increasing chemical formation (from surface warming by weakening transpiration and SOA radiative forcing). However, changes in precursor IPE as well as the LAI have limited impacts on surface $O_3$.

Sadiq et al. (2017) also showed positive $O_3$-vegetation feedback on the surface $O_3$ in a global model. Compared to their results, we show the strongest feedback in eastern China rather than western Europe, which is more reasonable, as the $O_3$ level in China is much higher than that in Europe (Lu et al., 2018). In addition, the effect of $O_3$-vegetation feedback on temperature is lower in our study. The fixed decoupled scheme in Sadiq et al. (2017) may have overestimated the effect of $O_3$ damage to stomatal conductance, leading to stronger feedback on $O_3$ concentrations and temperature. Furthermore, the



mechanisms of $O_3$ effects on IPE are different. Sadiq et al. (2017) showed increased IPE because of the warming feedback. However, such warming is not significant in our study (Fig. S6). Instead, we include direct effect of $O_3$ damage to IPE based on observations. Although the simulations show limited impacts of reduced IPE on surface $O_3$, the simultaneously reduced SOAs contribute to increased surface $O_3$ by weakening shortwave radiative forcing and increasing temperature in the eastern

U.S.

Our results are subject to uncertainties in modeled $O_3$ and damaging schemes. ModelE2-YIBs overestimates summer $O_3$, particularly in China (Fig. 1), which may exacerbate the damage to stomatal conductance and the consequent feedback. The $O_3$ damage parameterization by Sitch et al. (2007) is a semiphysical scheme that couples photosynthesis and stomatal

conductance. However, some observational studies have showed that the sluggish stomatal responses under chronic $O_3$ exposure lead to stomata losing function and decoupling from photosynthesis (Paoletti and Grulke, 2005; Gregg et al., 2006). The decoupled parameterization proposed by Lombardozzi et al. (2012) has been applied to estimate the the effect of $O_3$ damage to photosynthesis and stomatal conductance (Lombardozzi et al., 2015; Sadiq et al., 2017; Zhou et al., 2018). Nevertheless, we apply the parameterization by Sitch et al. (2007) because the damage is reasonably associated with ambient

$O_3$ level, and the scheme has been extensively evaluated against available observations (Yue et al., 2017; Yue and Unger, 2018). Fixed damage for low (even zero) $O_3$ included in some PFTs in the decoupled scheme may result in overestimation of $O_3$-vegetation feedback in the global model.

To our knowledge, this is the first time that the effect of $O_3$ damage to IPE is included in a fully coupled global chemistry-

carbon-climate model. Both the F and linear schemes can simulate reasonable reductions in IPE compared to global meta-analysis, although with large uncertainties. The reduced IPE, as precursors, have insignificant effects on surface $O_3$ concentrations in eastern China (Fig. 5 and Fig. 6), likely because of high anthropogenic emissions that undermine the feedback of IPE changes to surface $O_3$. However, the reduced IPE weakens SOA radiative forcing and increases surface temperature in the eastern U.S., where biogenic SOAs provide important contributions to total aerosols (Fine et al., 2008;

Goldstein et al., 2009). These results suggest that IPE feedback to the surface $O_3$ is quite uncertain and dependent on ambient precursors (anthropogenic vs. biogenic) and oxidizing capacity ($NO_x$-saturated vs. $NO_x$-limited).

Despite these uncertainties, our analyses highlight the importance of $O_3$–vegetation interactions in surface $O_3$ concentrations. The feedback should be considered in regional and global air quality models for more realistic simulations. Furthermore, the

effect of positive feedback on surface $O_3$ may potentially aggravate $O_3$ pollution in the future with increased ambient $O_3$ under a warming climate (Lei et al., 2012; Doherty et al., 2013).



**Data availability**

The observed hourly ozone concentrations for AQMN-MEE, CASTNET and EMEP were obtained from the Data Center of China's Ministry of Ecology and Environment (http://datacenter.mee.gov.cn/websjzx/queryIndex.vm), U.S. Environmental Protection Agency (https://java.epa.gov/castnet/clearsession.do) and EMEP Chemical Coordinating Centre

(https://www.emep.int/). The source codes for the ModelE2-YIBs are available through collaboration. Please submit a request to X. Yue (yuexu@nuist.edu.cn).

**Author contribution**

XY conceived the study. CG carried out the simulations and performed the analysis. YL and YM provided useful comments on the paper. CG, XY and HL prepared the manuscript with contributions from all coauthors.

**Competing interests**

The authors declare that they have no conflict of interest.

**Acknowledgments**

This work was supported by the National Natural Science Foundation of China (grant nos. 41975155, 91544219, 91744311, and 41475137) and National Key Research and Development Program of China (grant no. SQ2019YFA060013-02).

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



**Tables**

**Table 1. Summary of the seven experiments in ModelE2-YIBs.**

| Name | $O_3$ damage to photosynthesis | $O_3$ damage to stomatal conductance | $O_3$ damage to isoprene emissions |
|---|---|---|---|
| CTRL | None | None | Linear (offline) |
| DRY_high | F_high | F_high | F_high (offline) |
| DRY_ low | F_low | F_low | F_low (offline) |
| TOTAL_F_high | F_high | F_high | F_high |
| TOTAL_F_low | F_low | F_low | F_low |
| TOTAL_LINEAR_high | F_high | F_high | Linear |
| TOTAL_LINEAR_low | F_low | F_low | Linear |



**Figures**

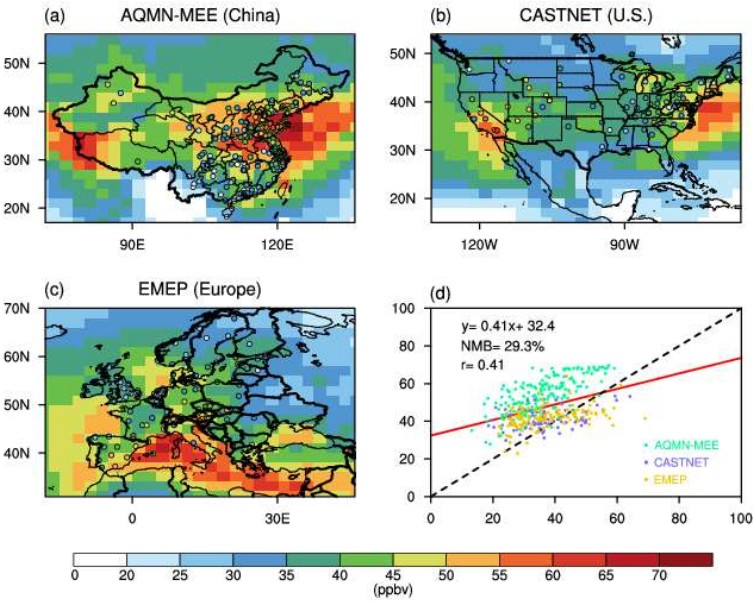

**Figure 1. Evaluations of simulated summer surface O₃ concentrations in the CTRL run. (a) - (c) Spatial distribution of observed O₃ concentrations (circle dots) in AQMN-MEE in China, CASTNET in the U.S. and EMEP in Europe, respectively, and the simulated**

5 **O₃ concentrations. (d) Scatter plots of O₃ concentrations (ppbv) over observational sites in the three regions. The X and Y axes indicate the observed and simulated O₃ concentrations, respectively. The red line shows the linear regression between the observed and simulated O₃ concentrations. The black dashed line shows the 1:1 lines.**


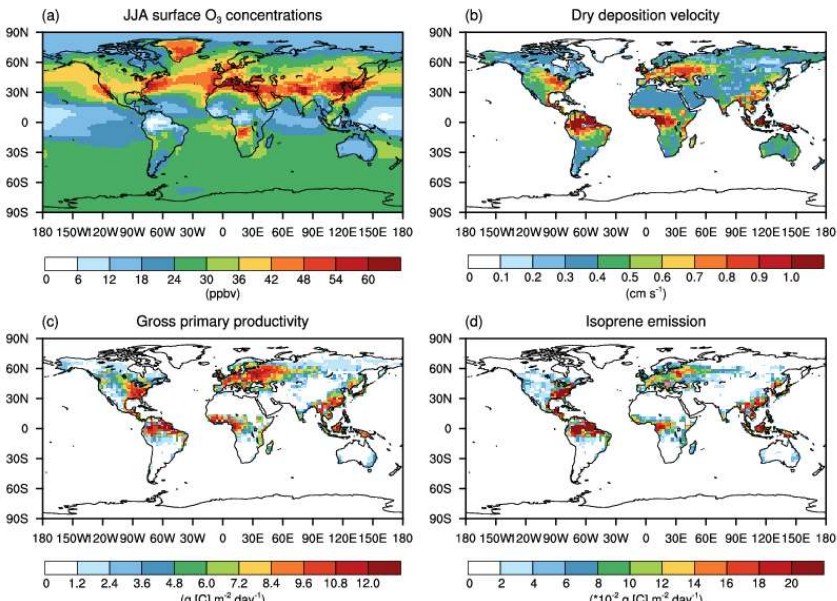

**Figure 2. The JJA-mean (a) surface O₃ concentrations, (b) O₃ dry deposition velocity, (c) gross primary productivity and (d) isoprene emissions in the CTRL simulation without O₃ damage to vegetation.**





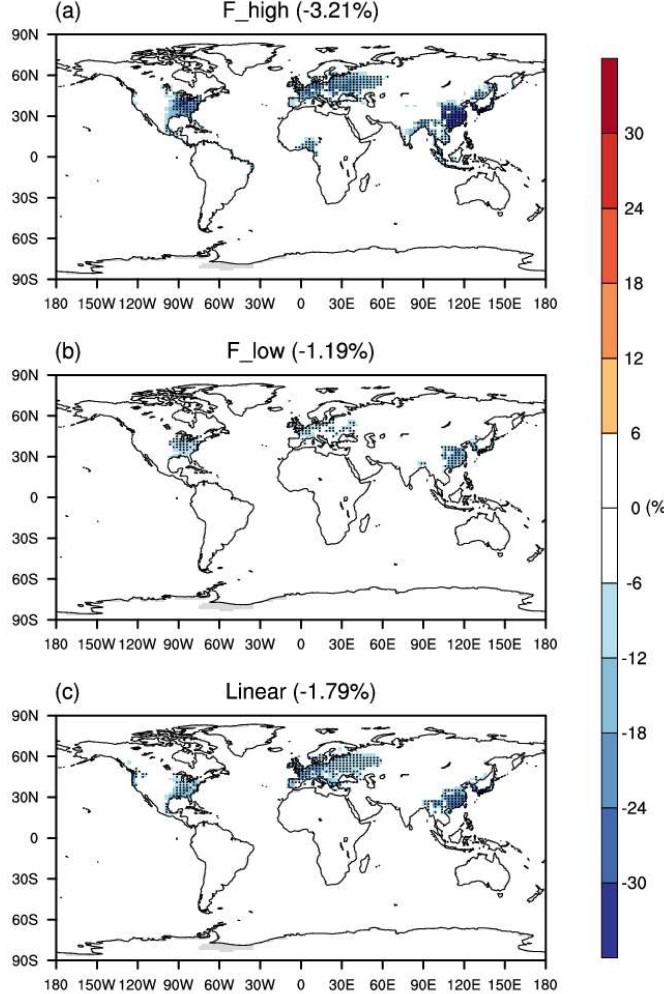

**Figure 3. Offline O$_3$ damage (%) to IPE averaged over summer using the F scheme with (a) high or (b) low sensitivities and results obtained by using the (c) linear scheme. The dotted grids shows significant damage at the 95% confidence level. Global land area-weighted percentage changes in IPE are shown in the titles.**



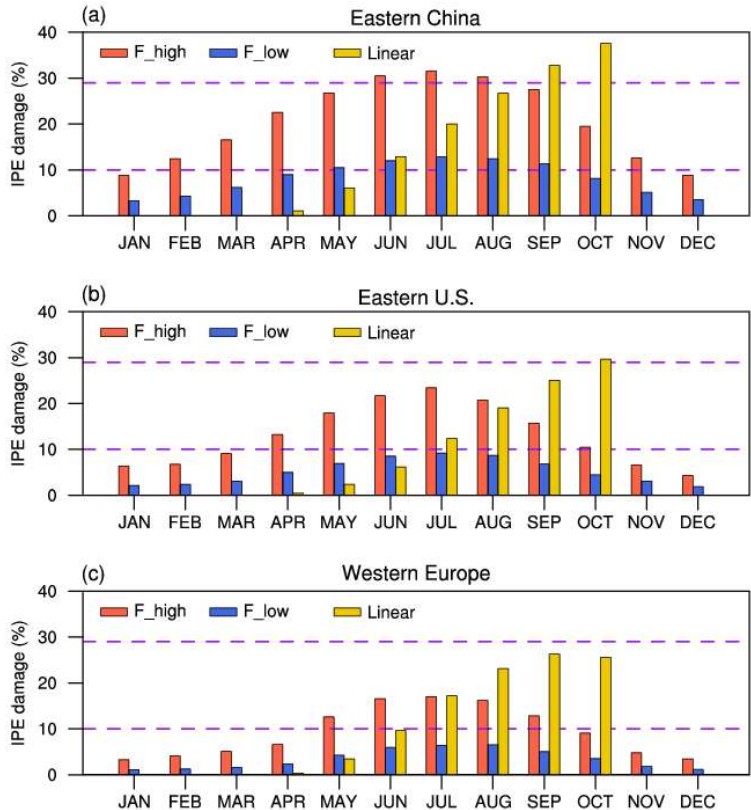

**Figure 4. Monthly mean percentage O₃ damage to IPE averaged over (a) eastern China, (b) the eastern U.S. and (c) western Europe by using the F scheme with high/low sensitivities and the linear scheme, respectively. The dashed lines indicate the range of IPE damage summarized by observational meta-analysis.**

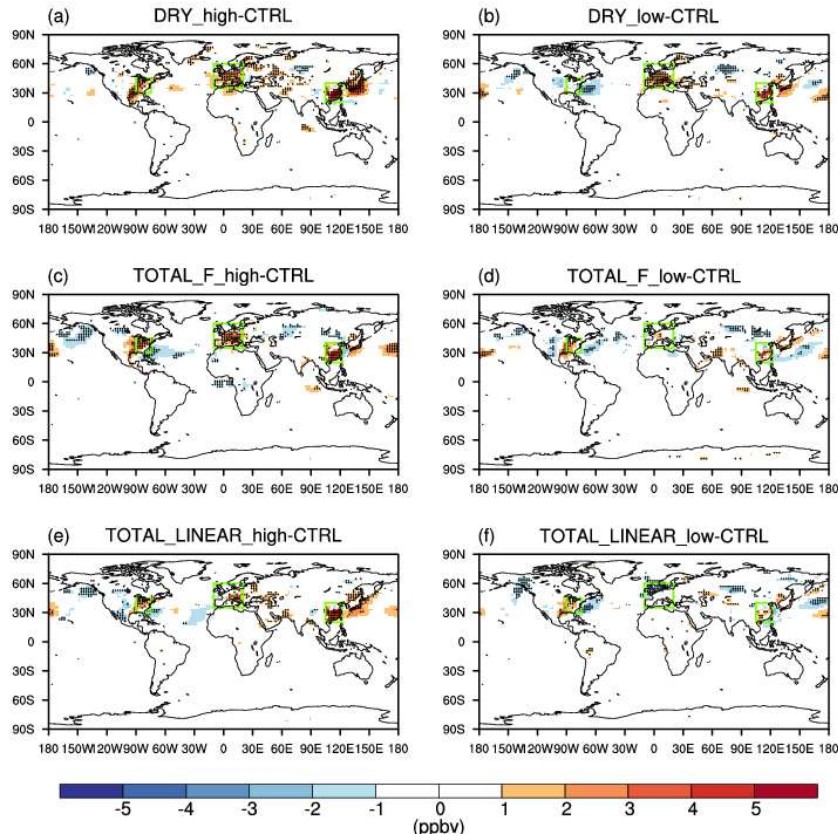

**Figure 5. O₃-vegetation feedback on surface O₃ concentrations during summer.** The results shown are changes in surface O₃ resulting from O₃ damage to stomatal conductance alone with (a) high and (b) low sensitivity. In addition to stomatal conductance, O₃ damage to IPE is also included by using the F scheme with (c) high and (d) low sensitivity. In comparison, O₃ damage to IPE is added for the linear scheme in (e) and (f). The dotted grids indicate significant changes at the 95% confidence level. The three box regions denote eastern China, the eastern U.S., and western Europe.





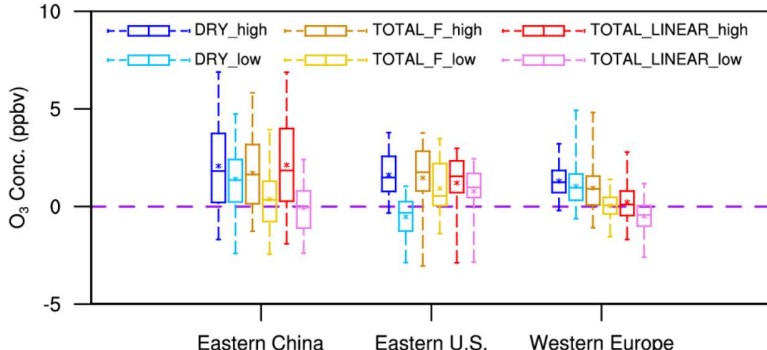

**Figure 6. Box plots of summer O₃ changes in three sensitive regions among different sensitivity experiments. The error bars show the ranges of O₃ changes in individual grids over the selected regions. Asterisks indicate the mean O₃ changes averaged over the selected regions.**

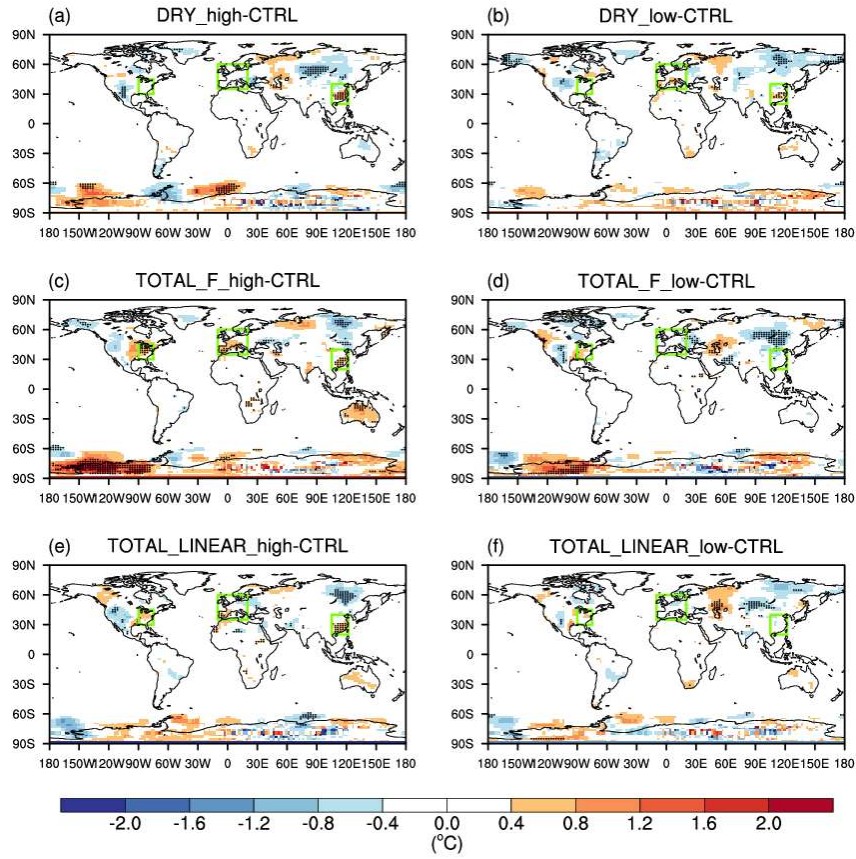

**Figure 7. Same as Fig. 5 but for changes in surface air temperature.**



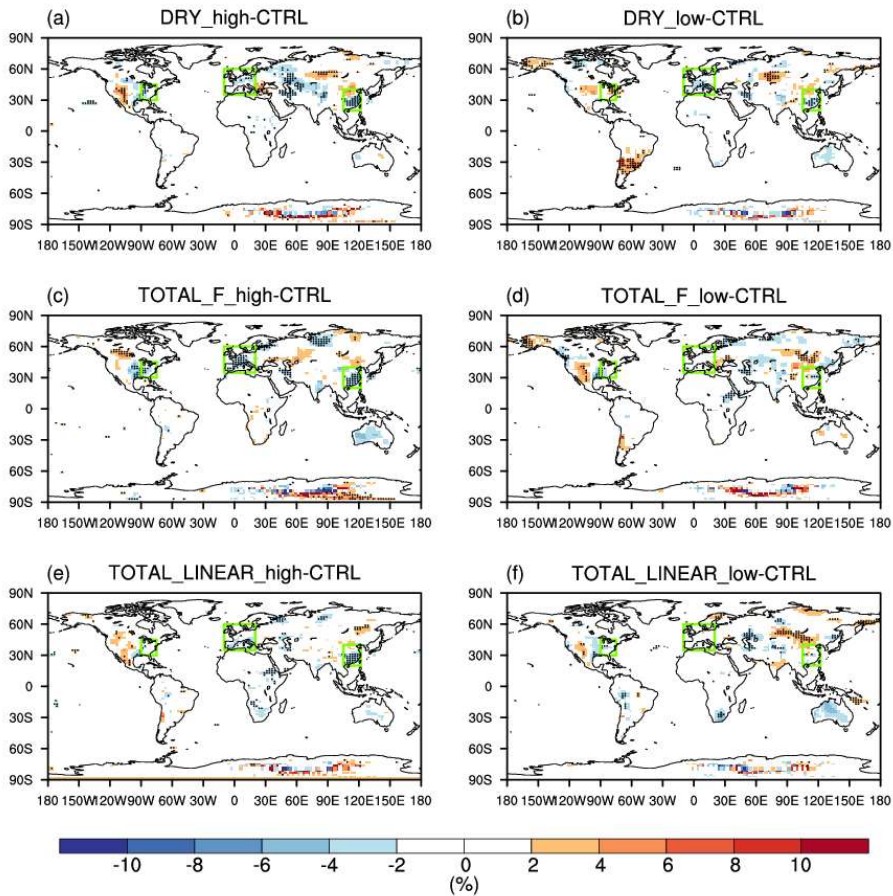

**Figure 8. Same as Fig. 5 but for changes in relatively humidity.**





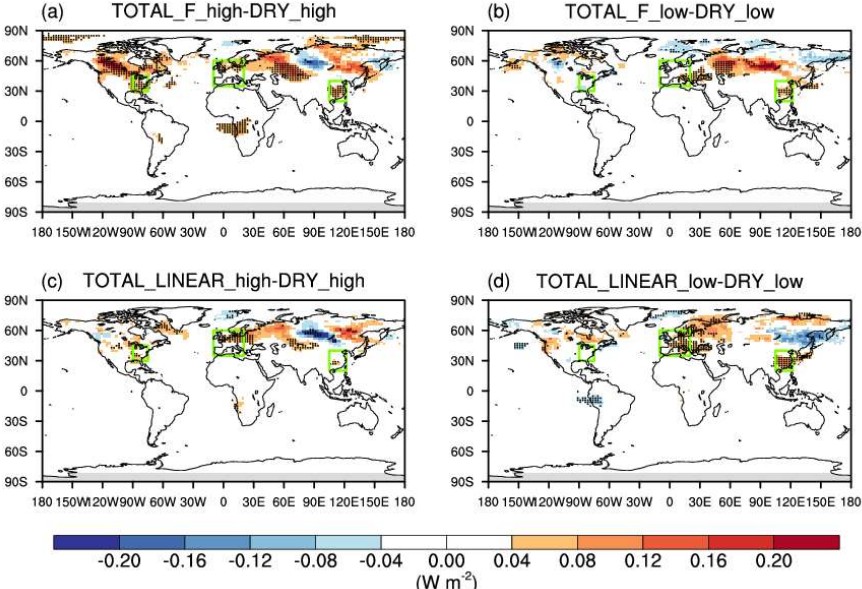

**Figure 9.** Effects of O₃-induced IPE reductions on SOA shortwave radiative forcing at the surface during the boreal summer. The impacts of O₃ damage to IPE are isolated by determining the differences in the experiments for (a) high and (b) low sensitivities by using the F schemes or the (c, d) linear scheme. Dotted grids indicate significant changes at the 95% confidence level.