# Peer review of "Ozone-vegetation feedback through dry deposition and isoprene emissions in a global chemistry-carbon-climate model"

_Atmospheric Chemistry and Physics, 2019_

## Referee Comment (RC1) · Anonymous Referee #1 · 14 Dec 2019

**Review of Gong et al. 2019**

Gong et al. present research using the NASA ModelE2-YIBs model to estimate the impact of ozone damage to vegetation on atmospheric composition. They implement a more detailed representation of ozone damage in a coupled land-atmosphere model and find that, in general, inhibition to stomatal conductance leads to ozone increases. Quantifying biosphere-atmosphere exchange processes such as ozone damage in coupled models is an important line of research, and this work will likely be fit for publication in ACP once the following comments are addressed.

**General Comments**

My major concern is with seemingly inconsistent results in various heavily vegetated areas, specifically Africa. In Figure 2, ozone concentrations in central Africa look to be ~48 ppbv in a region with a lot of vegetation. This is higher than ozone in other regions (e.g. North America) that do show ozone damage impacts. However, in Figures 3, 5, 7, and 8, there are no discernable ozone damage impacts shown in this area. Why is that the case? This is surprising and should be explained further in the manuscript.

**Specific Comments**

P2 L31: Citation needed for the statement that the majority of ozone deposition is through stomatal pathways.

P3: Despite the text critical of previous work, the authors here find a very similar ultimate impact of ozone damage on vegetation. This should be acknowledged here or elsewhere in the manuscript.

P4: A description of biogenic emissions is necessary in this section.

P6 Eq 10: What are the variables "n" and "i"?

P6 L28: What does "because of the data limit" mean?

P7: The CTRL statement as described here is confusing. The text states that damage is calculated offline using the Sitch et al. (2007) scheme, but Table 1. states "None". Which is it?

P7 L30: The linear fit in Figure 7d indicates and absolute bias of 32 ppbv. This should be acknowledged in the text as a limitation of this modeling approach.

P9 L 10: If the justification for focusing on northern hemispheric summer is that absolute changes to IPE are most significant during this time, why not show this in a figure instead of merely suggesting it?

P10 L10: The authors speculate that the changes are no due to IPE changes, but instead meteorology. This should be explained further in more detail or stated more clearly as speculation.

P11 L22: "likely related to the increased temperature…" further speculation. The sensitivity of the simulated ozone to temperature is not disentangled from other confounding factors. This should either be explicitly done, or the statement softened.

---

## Short Comment (SC1) · 8 Jan 2020

Does your model account for the influence of soil water availability on ozone deposition? A recent paper by Lin et al. (2019) demonstrated a key role for water availability in modulating O3 deposition variability on seasonal to interannual time scales via changes in stomatal conductance, with the effects on monthly mean daytime $V_{d,O3}$ variability as large as a factor of two. Their results are highly relevant to your literature review and discussions. It would be interesting to show if the effects of ozone damage on surface ozone discussed in your article differ substantially during dry versus wet years.

Reference:

[Figure]

Lin, Meiyun, Sergey Malyshev, Elena Shevliakova, Fabien Paulot, Larry W Horowitz, S Fares, T N Mikkelsen, and L Zhang, October 2019: Sensitivity of ozone dry deposition to ecosystem-atmosphere interactions: A critical appraisal of observations and simulations. Global Biogeochemical Cycles, 33(10), DOI:10.1029/2018GB006157.

This comment is posted by Meiyun Lin, Princeton University (https://www.gfdl.noaa.gov/meiyun-lin-homepage/)

---

## Referee Comment (RC2) · Anonymous Referee #2 · 18 Jan 2020

This study considers the impacts on surface ozone concentrations due to two ozone-vegetation feedback mechanisms, the dry deposition inhibition by ozone and the isoprene emission inhibition by ozone. This is an important scientific question that have been tackled by several previous studies. The unique aspect of this work is that the two feedback mechanisms are explicitly included in the ModelE2-YIBs model, and two levels of parameterized sensitivity were assessed for each of the two feedback mechanisms. The results show that the ozone-inhibition of dry deposition generally wins over the effects of ozone-inhibition of isoprene emissions, such that surface ozone increase over Eastern US, Europe, and Eastern China when the ozone effects are considered, relative to the control simulation (where no ozone effects are considered). In addition, indirect impacts on meteorology via weakened transpiration and enhanced solar radiation scattering by SOA also play a role.

Overall, I have a very favorable impression of this conceptual paper and consider it publishable after minor revisions. I do wish, however, that the authors can go beyond the common model validation methods and try to validate the model performance on the ozone-vegetation sensitivity. There are also key details about the model setup that needs to be included in the manuscript. See the comments below.

Major comments:

Section 2.1: What oxidants were considered from the two-product SOA production scheme? If ozone is one of the oxidants considered, is there significant feedback through this pathway (more O3 -> more SOA -> cooling -> reduced isoprene emission) ? The pathway that the authors described was (more isoprene -> more SOA -> cooling -> reduced isoprene emission)

Section 2.1: What assumptions were made regarding isoprene nitrate formation and its photochemical fate? This has long been shown to significantly impact the response of ozone to isoprene emissions.

The validation of the model performance in reproducing surface ozone concentration is unsatisfactory. The model, while no worse than others, does not reproduce well the ozone observations. More importantly, validating the mean surface ozone level does not really give insights to whether the model correctly (or better than other models) reproduces the ozone-vegetation relationship. I wish the authors can make an effort to go the extra mile and look at the ozone-temperature dependancy or the ozone-LAI dependency. Also, does the model perform better in the sensitivity simulations including vegetation-chemistry feedbacks?

The authors suggested that the reason for over-estimation of ozone over China was due to an overestimation of anthropogenic emissions? Is there justification for that?

How does the IPCC RCP8.5 emission (van Vuuren et al., 2011) compare to Chinese inventories. The authors also did not mention the basis of their isoprene emission. Have the authors validated their isoprene emissions for the three regions against inversion studies using satellite observations?

Minor comments:

Page 4, Lines 23-25: What is the scientific basis for parameterizing stomatal conductance as a function of these parameters, especially A_tot? I realize that a full answer to this question is beyond the scope of this study. Nevertheless, it might worthwhile to say a few words here or in the introduction to justify this assumption, which is central to the results of this study.

Page 4, Lines 25-26: missing reference for the canopy radiation scheme.

Page 5, line 12: 'online computed' should be 'computed online'

Page 5, line 27: How was F_O3 calculated and how was it related to g_s?

Page 6, line 23: What is n in Equation (10)?

Figure 1b: Please label the x and y axes. Also, the pastel colors in Figures 1b and S1 are extremely hard to see. Please consider changing the color scheme.

---

## Author Comment (AC1) · 11 Feb 2020

**Response to Comments of Reviewer #1**

**Manuscript number:** acp-2019-935

**Authors:** Cheng Gong, Yadong Lei, Yimian Ma, Xu Yue and Hong Liao

**Title:** Ozone-vegetation feedback through dry deposition and isoprene emissions in a global chemistry-carbon-climate model

*Gong et al. present research using the NASA ModelE2-YIBs model to estimate the impact of ozone damage to vegetation on atmospheric composition. They implement a more detailed representation of ozone damage in a coupled land-atmosphere model and find that, in general, inhibition to stomatal conductance leads to ozone increases. Quantifying biosphere-atmosphere exchange processes such as ozone damage in coupled models is an important line of research, and this work will likely be fit for publication in ACP once the following comments are addressed.*

**Response:**

Thank you for the helpful comments and suggestions. We have revised the manuscript carefully and the point-to-point responses are listed below.

**General comments:**

*My major concern is with seemingly inconsistent results in various heavily vegetated areas, specifically Africa. In Figure 2, ozone concentrations in central Africa look to be ~48 ppbv in a region with a lot of vegetation. This is higher than ozone in other regions (e.g. North America) that do show ozone damage impacts. However, in Figures 3, 5, 7, and 8, there are no discernable ozone damage impacts shown in this area. Why is that the case? This is surprising and should be explained further in the manuscript.*

**Response:**

Sorry for the confusion due to the low resolution of figures. If we zoom in Figure 2 on Africa (Figure R1), the 'heavily vegetated areas' (enclosed by the green rectangle), which is mainly covered by evergreen broadleaf forest (Figure R2), show low $O_3$ concentrations. Meanwhile, the region with high $O_3$ concentrations (enclosed by the blue rectangle) show quite low GPP and IPE. As a result, the weak $O_3$-vegetation interactions are reasonable in Africa.

In Sitch et al. (2007) schemes, different vegetation types show different performance to the $O_3$ exposure. Compared with evergreen or deciduous broadleaf forest, C3/C4 grassland and cropland have higher threshold $F_{O3,crit}$ (Supplementary Table S1). As a

result, C3/C4 grassland and cropland in African would suffer lower level of O₃ damage than the deciduous broadleaf forest in North America even under the similar level of O₃ exposure, leading to trivial ozone damage impacts in African in Figure 3, 5, 7, and 8.

[Figure]

Figure R1. The same as Figure 2 but zoomed in on Africa. The green box encloses the 'heavy vegetated areas' with high GPP and IPE. The blue box encloses areas where surface O₃ concentrations are high.

[Figure]

Figure R2. Land cover fraction of (a) C3 grassland , (b) evergreen broadleaf forest, (c) shrubland and (d) C4 grassland and cropland. The average cover of each PFT over the areas with high surface $O_3$ concentrations (blue box) is given in the subtitles.

**Specific Comments:**

*P2 L31: Citation needed for the statement that the majority of ozone deposition is through stomatal pathways.*

**Response:**

The sentence has been revised as:

'$O_3$ dry deposition is one of the important sink of tropospheric $O_3$ and mainly occurs over vegetation (Wesely, 1989). The stomatal uptake of vegetation plays an important role in this removal process. (Wesely and Hicks, 2000)' (Page 3, Lines 1-2)

*P3: Despite the text critical of previous work, the authors here find a very similar ultimate impact of ozone damage on vegetation. This should be acknowledged here or elsewhere in the manuscript.*

**Response:**

We have added the correspondingly statement in the second paragraph of Sect.4 Conclusion and discussion:

'Sadiq et al. (2017) also showed positive $O_3$-vegetation feedback on the surface $O_3$ in a global model. Compared to their results, we find an ultimate positive feedback with similar magnitude of surface $O_3$ concentrations but different spatial pattern. The strongest feedback in eastern China….' (Page 12, Lines 25-27)

*P4: A description of biogenic emissions is necessary in this section.*

**Response:**

The description of biogenic emissions has been added in the second paragraph in Sect. 2.1:

'…The LAI and tree growth are dynamically simulated with the allocation of carbon assimilation. The emissions of isoprene are calculated online as a function of Je photosynthesis (Eq. (1)), canopy temperature, intercellular $CO_2$, and $CO_2$ compensation

point (Arneth et al., 2007; Unger, 2013), and have been fully validated by Unger et al. (2013). Carbon fluxes, phenology, LAI, GPP, and net ecosystem exchange (NEE), ….' (Page 5, Lines 1-5)

*P6 Eq 10: What are the variables "n" and "i"?*

**Response:**

The Eq. (10) in origin manuscript and the correspondingly explanation has been revised as:

$$POD_1 = \int_1^n (F_{O_3} - 1)dt$$

(14)

'where $F_{O_3}$ is the $O_3$ uptake rate by stomata (nmol $O_3$ $m^{-2}$ $s^{-1}$), which is the same as that in Eq. (11). $dt$ indicates the time integration step and $n$ indicates the total number of time steps during the growing season.'   (Page 7, Lines 10-13)

*P6 L28: What does "because of the data limit" mean?*

**Response:**

As we described above, 'To date, only one study (Yuan et al., 2017) has explored the responses of IPE to different levels of $O_3$ damage for two poplar clones', so here we have to apply the PDI function for all vegetation types even though it is based on poplar observations. We have revised this sentence to clarify:

'Limited by the data availability, we apply the PDI function (Eq. (13)) for poplar to all vegetation types as follow:' (Page 7, Lines 16-17)

*P7: The CTRL statement as described here is confusing. The text states that damage is calculated offline using the Sitch et al. (2007) scheme, but Table 1. states "None". Which is it?*

**Response:**

Sorry for the confusion.

The CTRL run calculates offline ozone damaging, which does not feed back to affect vegetation growth and the stomatal uptake of ozone. As a result, we denote "None" for

this run in original Table 1. To clarify, we change "None" in Table 1 to "Offline". In text, we revised as follows:

'In the CTRL run, the effects of $O_3$ damage to photosynthesis, stomatal conductance, and IPE are calculated offline; such damages are not fed back to affect vegetation growth and dry deposition of $O_3$.'   (Page 7, Lines 27-29)

*P7 L30: The linear fit in Figure 7d indicates and absolute bias of 32 ppbv. This should be acknowledged in the text as a limitation of this modeling approach.*

**Response:**

The sentences have been revised as follow:

'Figure 1 shows a comparison of the simulated summer $O_3$ concentrations to the observations. The model in general captures reasonable spatial patterns with a correlation coefficient of 0.41. The NMBs between simulations and observations in U.S and Europe are 11.7% and 13.2%, respectively, which are comparable with the simulation performed by CESM (Lamarque et al., 2012; Sadiq et al., 2017). However, the model overestimates $O_3$ concentrations by 29.3% with a regression intercept of 32 ppbv, suggesting that simulated $O_3$ vegetation damage might be overestimated especially over some regions with low ambient $O_3$ level. The large overestimate is mainly a result of overestimation in China…' (Page 8, Lines 17-22)

*P9 L10: If the justification for focusing on northern hemispheric summer is that absolute changes to IPE are most significant during this time, why not show this in a figure instead of merely suggesting it?*

**Response:**

A new supplementary Figure S4 is added to show the absolute changes in IPE:

[Figure]

Supplementary Figure S4. Monthly mean absolute $O_3$ damage to IPE ($10^{-2}$ g[C] m$^{-2}$ day$^{-1}$) averaged over (a) eastern China, (b) the eastern U.S. and (c) western Europe by using the F scheme with high/low sensitivities and the linear scheme, respectively.

The main reason for focusing on boreal summer is that surface $O_3$ concentrations are high and vegetation grows vigorously in the northern hemisphere. Consequently, the $O_3$-vegetation-IPE interactions are supposed to be the strongest. In the text, we clarify as follows:

'…However, the IPE peaks during summer (Fig. S3), suggesting that absolute changes in IPE are most significant during this season (Fig. S4). Meanwhile, since the surface $O_3$ concentrations and the vegetation growth both peak during boreal summer in northern hemisphere, the $O_3$-vegetation interactions are supposed to be the strongest in this season. As a result, we focus our analyses on the summer to explore the $O_3$-vegetation interactions and feedback.' (Page 10, Lines 1-5)

*P10 L10: The authors speculate that the changes are no due to IPE changes, but instead meteorology. This should be explained further in more detail or stated more clearly as speculation.*

**Response:**

The sentence has been revised as follow:

'Nevertheless, inclusion of IPE reductions helps increase surface $O_3$ over the eastern U.S. (Figs. 5d/5f vs. Fig. 5b), which is unexpected since the reduction in IPE is supposed to decrease $O_3$ concentrations. These changes are speculated to be indirectly related to $O_3$-vegetation feedback to meteorology and would be further examined in the next section.' (Page 11, Lines 4-6)

*P11 L22: "likely related to the increased temperature…" further speculation. The sensitivity of the simulated ozone to temperature is not disentangled from other confounding factors. This should either be explicitly done, or the statement softened*

**Response:**

The sentence has been revised as follow:

'The increased temperature following reduced SOA concentrations are speculated as a possible cause for this result.' (Page 12, Lines 19-20)

**References**

Arneth, A., Niinemets, U., Pressley, S., Back, J., Hari, P., Karl, T., Noe, S., Prentice, I. C., Serca, D., Hickler, T., Wolf, A., and Smith, B.: Process-based estimates of terrestrial ecosystem isoprene emissions: incorporating the effects of a direct CO2-isoprene interaction, Atmospheric Chemistry and Physics, 7, 31-53, 10.5194/acp-7-31-2007, 2007.

Lombardozzi, D., Levis, S., Bonan, G., and Sparks, J. P.: Predicting photosynthesis and transpiration responses to ozone: decoupling modeled photosynthesis and stomatal conductance, Biogeosciences, 9, 3113-3130, 10.5194/bg-9-3113-2012, 2012

Sadiq, M., Tai, A. P. K., Lombardozzi, D., and Martin, M. V.: Effects of ozone-vegetation coupling on surface ozone air quality via biogeochemical and meteorological feedbacks, Atmospheric Chemistry and Physics, 17, 3055-3066, 10.5194/acp-17-3055-2017, 2017.

Sitch, S., Cox, P. M., Collins, W. J., and Huntingford, C.: Indirect radiative forcing of climate change through ozone effects on the land-carbon sink, Nature, 448, 791-U794, 10.1038/nature06059, 2007.

Unger, N.: Isoprene emission variability through the twentieth century, Journal of Geophysical Research-Atmospheres, 118, 13606-13613, 10.1002/2013jd020978, 2013.

Unger, N., Harper, K., Zheng, Y., Kiang, N. Y., Aleinov, I., Arneth, A., Schurgers, G., Amelynck, C., Goldstein, A., Guenther, A., Heinesch, B., Hewitt, C. N., Karl, T., Laffineur, Q., Langford, B., McKinney, K. A., Misztal, P., Potosnak, M., Rinne, J., Pressley, S., Schoon, N., and Seraca, D.: Photosynthesis-dependent isoprene emission from leaf to planet in a global carbon-chemistry-climate model, Atmospheric Chemistry and Physics, 13, 10243-10269, 10.5194/acp-13-10243-2013, 2013

Wesely, M. L.: PARAMETERIZATION OF SURFACE RESISTANCES TO GASEOUS DRY DEPOSITION IN REGIONAL-SCALE NUMERICAL-MODELS, Atmospheric Environment, 23, 1293-1304, 10.1016/0004-6981(89)90153-4, 1989.

Wesely, M. L., and Hicks, B. B.: A review of the current status of knowledge on dry deposition, Atmospheric Environment, 34, 2261-2282, 10.1016/s1352-2310(99)00467-7, 2000.

Yuan, X., Feng, Z., Liu, S., Shang, B., Li, P., Xu, Y., and Paoletti, E.: Concentration- and flux-based dose-responses of isoprene emission from poplar leaves and plants exposed to an ozone concentration gradient, Plant Cell and Environment, 40, 1960-1971, 10.1111/pce.13007, 2017

---

## Author Comment (AC2) · 11 Feb 2020

**Response to Short Comments from Meiyun Lin**

**Manuscript number:** acp-2019-935

**Authors:** Cheng Gong, Yadong Lei, Yimian Ma, Xu Yue and Hong Liao

**Title:** Ozone-vegetation feedback through dry deposition and isoprene emissions in a global chemistry-carbon-climate model

*Does your model account for the influence of soil water availability on ozone deposition? A recent paper by Lin et al. (2019) demonstrated a key role for water availability in modulating O3 deposition variability on seasonal to interannual time scales via changes in stomatal conductance, with the effects on monthly mean daytime Vd,O3 variability as large as a factor of two. Their results are highly relevant to your literature review and discussions. It would be interesting to show if the effects of ozone damage on surface ozone discussed in your article differ substantially during dry versus wet years.*

**Response:**

Thank you for the helpful comments. In ModelE2-YIBs, soil water stress is included following Porporato et al. (2001) to affect both plant photosynthesis and stomatal conductance (Yue and Unger, 2015). As a result, drought will reduce $O_3$ dry deposition and increase surface $[O_3]$. However, the model predicts present-day climate with small interannual variability, making it difficult to compare $O_3$ damaging responses in dry versus wet years. A CTM model driven with observed meteorology is a better tool to examine this issue.

Nevertheless, we agree that water availability is important for $O_3$-vegetation feedback. In the revised paper, we added following discussion:

'Variations in meteorological parameters may also influence $O_3$-vegetation feedback. Plant stomata tend to close under drought stress to prevent water loss. As a result, dry climate may weaken $O_3$-vegetation feedback through regulation of stomatal conductance (Lin et al., 2019). The effects of drought cannot be evaluated using ModelE2-YIBs, which simulates climatology with small interannual variability. In the future, a chemical transport model (CTM) coupled with a dynamic vegetation model (such as GC-YIBs developed by Lei et al. (2020)) will be used to examine drought impacts by using observation-based meteorological forcings.'

**References**

Lei, Y., Yue, X., Liao, H., Gong, C., and Zhang, L.: Implementation of Yale Interactive terrestrial Biosphere model v1.0 into GEOS-Chem v12.0.0: a tool for biosphere-chemistry interactions, Geosci. Model Dev., https://doi.org/10.5194/gmd-2019-281, in press, 2020.

Lin, M., Malyshev, S., Shevliakova, E., Paulot, F., Horowitz, L. W., Fares, S., Mikkelsen, T. N., and Zhang, L.: Sensitivity of Ozone Dry Deposition to Ecosystem-Atmosphere Interactions: A Critical Appraisal of Observations and Simulations, Global Biogeochemical Cycles, 33, 1264-1288, 10.1029/2018gb006157, 2019.

Porporato, A., Laio, F., Ridolfi, L., and Rodriguez-Iturbe, I.: Plants in water-controlled ecosystems: active role in hydrologic processes and response to water stress - III. Vegetation water stress, Advances in Water Resources, 24, 725-744, 10.1016/s0309-1708(01)00006-9, 2001.

Yue, X., and Unger, N.: The Yale Interactive terrestrial Biosphere model version 1.0: description, evaluation and implementation into NASA GISS ModelE2, Geoscientific Model Development, 8, 2399-2417, 10.5194/gmd-8-2399-2015, 2015.

---

## Author Comment (AC3) · 11 Feb 2020

**Response to Comments of Reviewer #2**

**Manuscript number:** acp-2019-935

**Authors:** Cheng Gong, Yadong Lei, Yimian Ma, Xu Yue and Hong Liao

**Title:** Ozone-vegetation feedback through dry deposition and isoprene emissions in a global chemistry-carbon-climate model

*This study considers the impacts on surface ozone concentrations due to two ozone vegetation feedback mechanisms, the dry deposition inhibition by ozone and the isoprene emission inhibition by ozone. This is an important scientific question that have been tackled by several previous studies. The unique aspect of this work is that the two feedback mechanisms are explicitly included in the ModelE2-YIBs model, and two levels of parameterized sensitivity were assessed for each of the two feedback mechanisms. The results show that the ozone-inhibition of dry deposition generally wins over the effects of ozone-inhibition of isoprene emissions, such that surface ozone increase over Eastern US, Europe, and Eastern China when the ozone effects are considered, relative to the control simulation (where no ozone effects are considered). In addition, indirect impacts on meteorology via weakened transpiration and enhanced solar radiation scattering by SOA also play a role.*

*Overall, I have a very favorable impression of this conceptual paper and consider it publishable after minor revisions. I do wish, however, that the authors can go beyond the common model validation methods and try to validate the model performance on the ozone-vegetation sensitivity. There are also key details about the model setup that needs to be included in the manuscript. See the comments below.*

**Response:**

Thank you for the helpful comments and suggestions. We have revised the manuscript carefully and the point-to-point responses are listed below.

*Major comments:*

*Section 2.1: What oxidants were considered from the two-product SOA production scheme? If ozone is one of the oxidants considered, is there significant feedback through this pathway (more O3 -> more SOA -> cooling -> reduced isoprene emission) ? The pathway that the authors described was (more isoprene -> more SOA -> cooling -> reduced isoprene emission)*

**Response:**

For the two-product SOA production scheme applied in ModelE2-YIBs, $O_3$ is the only oxidant that considered.

We further examine the feedback of 'more $O_3$ -> more SOA -> cooling'. Since $O_3$ concentrations are significantly enhanced (more $O_3$) when considering the effect of $O_3$ damage to photosynthesis and stomatal conductance (Fig. 5a and 5b), differences of SOA shortwave radiative forcing between DRY_high or DRY_low and CTRL experiments can be utilized to check whether SOA increases with more $O_3$. As is shown in Fig. R1, the SOA forcing shows very limited changes in eastern China, eastern America, and western Europe, where large $O_3$ enhancements are predicted (Fig. 5a and 5b). Such magnitude is much smaller than that in Fig. 9, which stands for the other pathway (more isoprene -> more SOA -> cooling). As a result, the weaker SOA cooling effect is driven by damaged IPE rather than the enhanced $O_3$ concentrations.

Meanwhile, we did not consider the feedback of SOA cooling on isoprene emissions. Instead, we speculated that weaker SOA cooling (less SOA) promoted temperature and surface $O_3$ concentrations in eastern U.S.

[Figure]

Figure R1. Effects of (a) high and (b) low $O_3$ vegetation damages on SOA shortwave radiative forcing at the surface during the boreal summer. Dotted grids indicate significant changes at the 95% confidence level. Eastern China, eastern U.S. and western Europe are enclosed by green rectangles.

*Section 2.1: What assumptions were made regarding isoprene nitrate formation and its photochemical fate? This has long been shown to significantly impact the response of ozone to isoprene emissions.*

**Response:**

Only three chemical reactions are considered in ModelE2-YIBs related to isoprene:

$$C_5H_8 + OH \rightarrow HCHO + Alkenes$$

$$C_5H_8 + O_3 \rightarrow HCHO + Alkenes$$

$$C_5H_8 + NO_3 \rightarrow HO_2 + Alkenes$$

Both HCHO and $HO_2$ further contribute to the formation of ozone.

The last paragraph of Sect. 2.1 has been revised as:

'Isoprene and α-pinene are considered as the precursors for biogenic secondary organic aerosols (SOA) in ModelE2-YIBs, which are computed online based on the two-product scheme developed by Chung and Seinfeld (2002). Isoprene can be oxidized by $O_3$ as follows:

$$C_5H_8 + O_3 \rightarrow HCHO + A_1P_1 + A_2P_2$$
(5)

Changes for semivolatile product $P_i$ (i=1,2) at each time step (*dt*) are calculated by:

$$\frac{dP_i}{dt} = A_i * rr * [O_3] * [C_5H_8]$$
(6)

where rr is the chemical reaction rate of $O_3$ and isoprene calculated by Arrhenius equation. $[O_3]$ and $[C_5H_8]$ are the $O_3$ and isoprene concentrations, respectively. $A_i$ is the molar based stoicheiometric coefficient depending on SOA formation pathways (high or low $NO_x$) (Lane et al., 2008). Temperature (T) dependence on partitioning coefficient (Kp) for P1 and P2 are given by the Clausius-Clapeyron equation:

$$K_p = K_{sc} \frac{T}{T_{sc}} \exp\left[\frac{\Delta H}{R}\left(\frac{1}{T} - \frac{1}{T_{sc}}\right)\right]$$
(7)

where ΔH is the enthalpy of vaporization and is set as 42.0 kJ mol[-1] for isoprene (Chung and Seinfeld, 2002;Henze and Seinfeld, 2006) and 72.9 kJ mol[-1] for α-pinene. Ksc is the saturation concentrations at the temperature Tsc (295 K) and set as 1.62 (0.064) m[3] μg[-1] and 0.0086 (0.0026) m[3] μg[-1] for the two products formed by oxidation of isoprene (α-pinene), respectively (Presto et al., 2005;Henze and Seinfeld, 2006).' (Page5 Lines 21-31; Page 6 Lines 1-4)

*The validation of the model performance in reproducing surface ozone concentration is unsatisfactory. The model, while no worse than others, does not reproduce well the ozone observations. More importantly, validating the mean surface ozone level does not really give insights to whether the model correctly (or better than other models) reproduces the ozone-vegetation relationship. I wish the authors can make an effort to*

*go the extra mile and look at the ozone-temperature dependancy or the ozoneLAI dependency. Also, does the model perform better in the sensitivity simulations including vegetation-chemistry feedbacks?*

**Response:**

Simulated $O_3$ concentrations do show certain biases compared to surface observations. However, if we validate maximum daily 8-hour average (MDA8) $[O_3]$, we found that the model shows much lower biases (Fig. S1 in the revised manuscript). The main reason for the overestimation is that the model predicts high nighttime $[O_3]$ that are not consistent with observations. Since $O_3$-vegetation interactions usually occur in the daytime, the updated validation shows that ModelE2 is good to use for this study.

[Figure]

Supplementary Fig. S1. Scatter plots of (a) daily mean and (b) MDA8 $O_3$ concentrations (ppbv) over observational sites in China. The purple line shows the linear regression between the observed and simulated $O_3$ concentrations. The black dashed line shows the 1:1 lines.

Ozone-vegetation relationships have been fully evaluated in our previous studies. For example, we validated $O_3$-GPP relations for six main vegetation types in Yue and Unger (2018) as follows:

[Figure]

**Figure R2.** Percentage changes in GPP for six main plant functional types (PFTs) caused by $O_3$. Red points on each panel represent literature-based measurements. The linear regression is denoted as a red solid line, with 95% confidence intervals shown as dashed lines. Blue points represent simulated GPP changes from offline sensitivity experiments (Methods), with error bars indicating the range of prediction from low to high $O_3$ damaging sensitivities. The slopes of observed ($S_o$, mean ± 95% confidence interval) and modeled ($S_m$, mean ± (high-low)/2 sensitivity) GPP-$O_3$ sensitivity is shown on each panel (figure from Yue, X., and Unger, N.: Fire air pollution reduces global terrestrial productivity, Nature Communications, 9, 5413, 2018).

We validated $O_3$ damages to stomatal conductance for deciduous trees in Yue et al. (2016) as follows:

[Figure]

**Figure R3.** Percentage changes in (a) photosynthesis and (b) stomatal conductance averaged across 20 deciduous broadleaf forest flux tower sites in response to different levels of [$O_3$]. The derived percentage changes (including uncertainties) based on the fits are plotted against observations for (c) photosynthesis and (d) stomatal conductance (figure from Yue, X., Keenan, T. F., Munger, W., and Unger, N.: Limited effect of ozone reductions on the 20-year photosynthesis trend at Harvard forest, Global Change Biology, 22, 3750-3759, 2016).

The relations between $O_3$ and LAI can not be evaluated as such observations are not available. However, based on good performance in simulating $O_3$-GPP and GPP-LAI (Yue and Unger, 2015) relationships, we consider ModelE2-YIBs model is appropriate to use for exploring $O_3$-vegetation interactions.

Finally, inclusion of $O_3$-vegetation feedback does not necessarily improve the model performance. The main purpose for this study is to explore the processes and magnitude

of $O_3$-vegetation feedback. The positive feedback we revealed may further enhance the model biases, suggesting that additional efforts are required to reduce modeling uncertainties in surface $O_3$.

*The authors suggested that the reason for over-estimation of ozone over China was due to an overestimation of anthropogenic emissions? Is there justification for that? How does the IPCC RCP8.5 emission (van Vuuren et al., 2011) compare to Chinese inventories. The authors also did not mention the basis of their isoprene emission. Have the authors validated their isoprene emissions for the three regions against inversion studies using satellite observations?*

**Response:**

We have revised the first paragraph in Sect. 3.1 as follow:

'Figure 1 shows a comparison of the simulated summer $O_3$ concentrations to the observations. The model in general captures reasonable spatial patterns with a correlation coefficient of 0.41. The NMBs between simulations and observations in U.S and Europe are 11.7% and 13.2%, respectively, which are comparable with the simulation performed by CESM (Lamarque et al., 2012; Sadiq et al., 2017). However, the model overestimates $O_3$ concentrations by 29.3% with a regression intercept of 32 ppbv, suggesting that simulated $O_3$ vegetation damage might be overestimated especially over some regions with low ambient $O_3$ level. The large overestimate is mainly a result of overestimation in China. However, if we validate maximum daily 8-hour average (MDA8) $O_3$ concentrations, we found that the model shows much lower biases (Fig. S1). The main reason for the overestimation is that the model predicts high nighttime $O_3$ concentrations that are not consistent with observations. Since $O_3$-vegetation interactions usually occur in the daytime, the validation shows that ModelE2-YIBs is good to use for this study. Meanwhile, most of the observational sites in AQMN-MEE are located in urban area, which might be another reason for the surface $O_3$ overestimates in China (Yue et al., 2017).' (Page 8, Lines 17-27)

As for the isoprene emissions, extensive validation has been done in previous study (Unger et al., 2013). They showed that a control simulation reproduced 50% of the variability across different ecosystems and seasons in a global database of 28 measured campaign-average fluxes, and captured the observed variance in the 30 min average diurnal cycle ($R^2$ =64–96%) at nine sites. The description of isoprene emissions has been added in the second paragraph in Sect. 2.1:

'…The LAI and tree growth are dynamically simulated with the allocation of carbon assimilation. The emissions of isoprene are calculated online as a function of Je photosynthesis (Eq. 1), canopy temperature, intercellular $CO_2$, and $CO_2$ compensation point (Arneth et al., 2007;Unger, 2013), and have been fully validated by Unger et al. (2013). Carbon fluxes, phenology, LAI, GPP, and net ecosystem exchange (NEE), ….' (Page 5, Lines 1-5)

*Minor comments:*

*Page 4, Lines 23-25: What is the scientific basis for parameterizing stomatal conductance as a function of these parameters, especially A_tot? I realize that a full answer to this question is beyond the scope of this study. Nevertheless, it might worthwhile to say a few words here or in the introduction to justify this assumption, which is central to the results of this study.*

**Response:**

Plant photosynthesis is closely related to stomatal conductance. The higher A_tot requires larger Gs to allow more $CO_2$ enter the leaves for photosynthesis. Such relationship has been parameterized by the Farquhar and Ball-Berry models, which has been widely utilized in land ecosystem simulation (e.g. Farquhar et al., 1980;Ball et al., 1987;Sitch et al., 2007;Bonan et al., 2011;Lombardozzi et al., 2012;Yue and Unger, 2015;Deryng et al., 2016;Sadiq et al., 2017).

The second paragraph of Sect. 2.1 has been revised as:

'The YIBs model is a dynamic vegetation model that includes 9 plant functional types (PFTs) (Table S1) and can simulate biophysical processes of photosynthesis, transpiration and respiration with variations in meteorological fields. **Since the higher leaf photosynthesis requires larger stomatal conductance to allow more $CO_2$ enter the leaves, leaf photosynthesis and stomatal conductance are closely related** and calculated using the Farquhar and Ball–Berry models **(Farquhar et al., 1980;Ball et al., 1987)** as follows:

$A_{tot} = min(J_c, J_e, J_s)$

(1)

$g_s = m \frac{(A_{tot} - R_d) \times RH}{c_s} + b$

(2)

where the total leaf photosynthesis ($A_{tot}$) is the minimum value of the ribulose-1,5-bisphosphate carboxylase (RuBisCO)-limited rate of carboxylation ($J_c$), light-limited rate ($J_e$), and export-limited rate ($J_s$). **Stomatal conductance for $H_2O$ ($g_s$)** is calculated

by the $A_{tot}$, dark respiration rate ($R_d$), relative humidity ($RH$) and $CO_2$ concentration at the leaf surface ($c_s$). The values of $m$ and $b$ are different for different PFTs (Table S1). A canopy radiation scheme is applied in YIBs to separate diffuse and direct light for sunlit and shaded leaves **(Spitters et al., 1986)**. The LAI and tree growth are dynamically simulated with the allocation of carbon assimilation. Carbon fluxes, phenology, LAI, GPP, and net ecosystem exchange (NEE), as well as other parameters of vegetation in ModelE2-YIBs, have been previously extensively evaluated and agree well with the observations (Yue and Unger, 2015). **In addition, ModelE2-YIBs shows good performance in simulating O₃-vegetation interactions such as O₃-GPP and O₃-gₛ relationships (Yue et al., 2016; Yue et al., 2018).**' (Page 4 Lines 21-31; Page 5 Lines 1-7)

*Page 4, Lines 25-26: missing reference for the canopy radiation scheme.*

**Response:**

Revised.

*Page 5, line 12: 'online computed' should be 'computed online'*

**Response:**

Revised.

*Page 5, line 27: How was F_O3 calculated and how was it related to g_s?*

**Response:**

The equation for $F_{O_3}$ calculation has been added as follow:

'A semi-mechanistic scheme proposed by Sitch et al. (2007) is applied in this study that simulates the effect of $O_3$ damage to the photosynthesis rate and stomatal conductance via the following formulas:

$$A_{totd} = F \times A_{tot} \tag{8}$$
$$g_{sd} = F \times g_s \tag{9}$$

where $A_{totd}$ ($g_{sd}$) and $A_{tot}$ ($g_s$) are the O₃-affected and original total leaf photosynthesis (stomatal conductance), respectively. $F$ is the ratio between affected and original photosynthesis. It depends on the instantaneous leaf uptake of $O_3$ as follows:

$$F = 1 - a \times max\,[F_{O_3} - F_{O_3,crit}, 0.0] \qquad (10)$$

where parameter *a* represents the $O_3$ damaging sensitivity dependent on vegetation types with a range from low to high values. $F_{O3,crit}$ is a critical threshold for damage (Table S1). $F_{O3}$ is the $O_3$ uptake rate by the stomata, which is calculated by:

$$F_{O3} = \frac{[O_3]}{R_a + [\frac{k_{O3}}{g_{sd}}]} \qquad (11)$$

Where $[O_3]$ is the surface $O_3$ concentrations and $R_a$ is the aerodynamic resistance in Eq. (3). $k_{O3}$ is 1.67, which is the ratio of leaf resistance for $O_3$ to leaf resistance for water vapor. This scheme has been used to explore $O_3$ damages to vegetation in many previous studies…..' (Page 6 Lines 7-19)

*Page 6, line 23: What is n in Equation (10)?*

**Response:**

The Eq. (10) in the origin manuscript and the correspondingly explanation has been revised as:

$$POD_1 = \int_1^n (F_{O_3} - 1)dt$$

(14)

'where $F_{O3}$ is the $O_3$ uptake rate by stomata (nmol $O_3$ m$^{-2}$ s$^{-1}$), which is the same as that in Eq. (11). *dt* indicates the time integration step and *n* indicates the total number of time steps during the growing season.' (Page 7 Lines 10-12)

*Figure 1b: Please label the x and y axes. Also, the pastel colors in Figures 1b and S1 are extremely hard to see. Please consider changing the color scheme.*

**Response:**

Revised.

Figure 1:

[Figure]

Figure S1 (Figure S2 in the revised manuscript):

[Figure]

**References**

Arneth, A., Niinemets, U., Pressley, S., Back, J., Hari, P., Karl, T., Noe, S., Prentice, I. C., Serca, D., Hickler, T., Wolf, A., and Smith, B.: Process-based estimates of terrestrial ecosystem isoprene emissions: incorporating the effects of a direct CO2-isoprene interaction, Atmospheric Chemistry and Physics, 7, 31-53, 10.5194/acp-7-31-2007, 2007.

Ball, J. T., Woodrow, I. E., and Berry, J. A.: A model predicting stomatal conductance and its contribution to the control of photosynthesis under different environmental conditions, Progress in Photosynthesis Research, 4, 221-224, 1987.

Bonan, G. B., Lawrence, P. J., Oleson, K. W., Levis, S., Jung, M., Reichstein, M., Lawrence, D. M., and Swenson, S. C.: Improving canopy processes in the Community Land Model version 4 (CLM4) using global flux fields empirically inferred from FLUXNET data, Journal of Geophysical Research-Biogeosciences, 116, 10.1029/2010jg001593, 2011.

Chung, S. H., and Seinfeld, J. H.: Global distribution and climate forcing of carbonaceous aerosols, Journal of Geophysical Research-Atmospheres, 107, 10.1029/2001jd001397, 2002.

Deryng, D., Elliott, J., Folberth, C., Mueller, C., Pugh, T. A. M., Boote, K. J., Conway, D., Ruane, A. C., Gerten, D., Jones, J. W., Khabarov, N., Olin, S., Schapho, S., Schmid, E., Yang, H., and Rosenzweig, C.: Regional disparities in the beneficial effects of rising CO2 concentrations on crop water productivity, Nature Climate Change, 6, 786-+, 10.1038/nclimate2995, 2016.

Farquhar, G. D., Caemmerer, S. V., and Berry, J. A.: A BIOCHEMICAL-MODEL OF PHOTOSYNTHETIC CO2 ASSIMILATION IN LEAVES OF C-3 SPECIES, Planta, 149, 78-90, 10.1007/bf00386231, 1980.

Henze, D. K., and Seinfeld, J. H.: Global secondary organic aerosol from isoprene oxidation, Geophysical Research Letters, 33, 10.1029/2006gl025976, 2006.

Lane, T. E., Donahue, N. M., and Pandis, S. N.: Effect of NOx on secondary organic aerosol concentrations, Environmental Science & Technology, 42, 6022-6027, 10.1021/es703225a, 2008.

Lamarque, J. F., Emmons, L. K., Hess, P. G., Kinnison, D. E., Tilmes, S., Vitt, F., Heald, C. L., Holland, E. A., Lauritzen, P. H., Neu, J., Orlando, J. J., Rasch, P. J., and Tyndall, G. K.: CAM-chem: description and evaluation of interactive atmospheric chemistry in the Community Earth System Model, Geoscientific Model Development, 5, 369-411, 10.5194/gmd-5-369-2012, 2012.

Lombardozzi, D., Sparks, J. P., Bonan, G., and Levis, S.: Ozone exposure causes a decoupling of conductance and photosynthesis: implications for the Ball-Berry stomatal conductance model, Oecologia, 169, 651-659, 10.1007/s00442-011-

2242-3, 2012.

Presto, A. A., Hartz, K. E. H., and Donahue, N. M.: Secondary organic aerosol production from terpene ozonolysis. 2. Effect of NOx concentration, Environmental Science & Technology, 39, 7046-7054, 10.1021/es050400s, 2005.

Sadiq, M., Tai, A. P. K., Lombardozzi, D., and Martin, M. V.: Effects of ozone-vegetation coupling on surface ozone air quality via biogeochemical and meteorological feedbacks, Atmospheric Chemistry and Physics, 17, 3055-3066, 10.5194/acp-17-3055-2017, 2017.

Sitch, S., Cox, P. M., Collins, W. J., and Huntingford, C.: Indirect radiative forcing of climate change through ozone effects on the land-carbon sink, Nature, 448, 791-U794, 10.1038/nature06059, 2007.

Spitters, C. J. T., Toussaint, H., and Goudriaan, J.: SEPARATING THE DIFFUSE AND DIRECT COMPONENT OF GLOBAL RADIATION AND ITS IMPLICATIONS FOR MODELING CANOPY PHOTOSYNTHESIS .1. COMPONENTS OF INCOMING RADIATION, Agricultural and Forest Meteorology, 38, 217-229, 10.1016/0168-1923(86)90060-2, 1986.

Unger, N.: Isoprene emission variability through the twentieth century, Journal of Geophysical Research-Atmospheres, 118, 13606-13613, 10.1002/2013jd020978, 2013.

Unger, N., Harper, K., Zheng, Y., Kiang, N. Y., Aleinov, I., Arneth, A., Schurgers, G., Amelynck, C., Goldstein, A., Guenther, A., Heinesch, B., Hewitt, C. N., Karl, T., Laffineur, Q., Langford, B., McKinney, K. A., Misztal, P., Potosnak, M., Rinne, J., Pressley, S., Schoon, N., and Seraca, D.: Photosynthesis-dependent isoprene emission from leaf to planet in a global carbon-chemistry-climate model, Atmospheric Chemistry and Physics, 13, 10243-10269, 10.5194/acp-13-10243-2013, 2013.

Yue, X., and Unger, N.: The Yale Interactive terrestrial Biosphere model version 1.0: description, evaluation and implementation into NASA GISS ModelE2, Geoscientific Model Development, 8, 2399-2417, 10.5194/gmd-8-2399-2015, 2015.

Yue, X., Keenan, T. F., Munger, W., and Unger, N.: Limited effect of ozone reductions on the 20-year photosynthesis trend at Harvard forest, Global Change Biology, 22, 3750-3759, 10.1111/gcb.13300, 2016.

Yue, X., and Unger, N.: Fire air pollution reduces global terrestrial productivity, Nature Communications, 9, 5413, 10.1038/s41467-018-07921-4, 2018.

---

## Author Response (AR2)

**Response to Comments of Editor**

**Manuscript number:** acp-2019-935

**Authors:** Cheng Gong, Yadong Lei, Yimian Ma, Xu Yue and Hong Liao

**Title:** Ozone-vegetation feedback through dry deposition and isoprene emissions in a global chemistry-carbon-climate model

*I decided to accept your publication, subject to minor revisions.*

**Response:**

Thank you for the helpful comments and suggestions. We have revised the manuscript and the point-to-point responses are listed below. Please notice that there are some minor changes in the numbers in main text after the inclusion of new Table 2, which derives average changes for all grids in specific regions. Our previous statistics is derived based only on grids with significant changes.

*1) First of all I suggest to summarize your results per experiment and region in a Table, to aid the synthesis of results.*

**Response:**

A new Table 2 has been added to summarize the results.

**Table 2.** Summary of the $O_3$-vegetation feedbacks on summertime (June-August) mean surface $O_3$ concentrations ($[O_3]$), surface air temperature (T), surface relative humidity (RH), and isoprene emissions (IPE) in different sensitivity experiments. The values are calculated as the online differences between sensitivity and CTRL experiments. At each region, the minimum and maximum changes are shown as uncertainties.

| Experiments | Regions | $\Delta[O_3]$ (ppbv) | $\Delta T$ (°C) | $\Delta RH$ (%) | $\Delta IPE$ ($10^{-3}$ g[C] m$^{-2}$ day$^{-1}$) |
|---|---|---|---|---|---|
| DRY_high | Eastern China | 2.1 [-2.1, 7.4] | 0.3 [-0.7, 1.0] | -1.1 [-5.8, 5.4] | -0.1 [-1.9, 2.1] |
| | Eastern U.S. | 1.8 [-0.6, 4.0] | 0.07 [-0.2, 0.3] | -1.0 [-4.4, 2.0] | 0.3 [-0.8, 4.9] |
| | Western Europe | 1.3 [-0.5, 3.8] | -0.05 [-0.8, 0.3] | -0.8 [-3.4, 2.6] | -0.02 [-1.8, 1.8] |
| DRY_low | Eastern China | 1.2 [-2.3, 4.6] | 0.1 [-0.5, 0.7] | -0.5 [-4.0, 4.5] | 0.04 [-1.8, 1.7] |
| | Eastern U.S. | -0.3 [-2.7, 1.8] | 0.1 [-0.1, 0.3] | 1.5 [-1.3, 4.5] | 0.8 [-0.7, 5.4] |
| | Western Europe | 1.0 [-0.8, 5.3] | 0.07 [-0.5, 1.0] | -1.0 [-6.7, 1.7] | -0.1[-3.5, 0.8] |

| | | | | | |
|---|---|---|---|---|---|
| TOTAL_F_high | Eastern China | 1.5 [-2.1, 5.4] | 0.2 [-0.5, 0.8] | -2.0 [-5.7, 1.8] | -2.3 [-6.8, 0.3] |
| | Eastern U.S. | 1.4 [-3.1, 4.3] | 0.5 [-0.2, 0.9] | -0.9 [-4.7, 1.9] | -2.6 [-7.0, 0.2] |
| | Western Europe | 1.2 [-1.2, 5.0] | 0.2 [-0.4, 1.0] | -1.1 [-6.5, 1.8] | -0.8 [-4.5, 0.7] |
| TOTAL_F_low | Eastern China | 0.02 [-3.3, 3.6] | -0.1 [-0.9, 0.3] | 0.05 [-3.1, 5.4] | -1.0 [-3.3, 2.5] |
| | Eastern U.S. | 1.6 [-0.4, 4.5] | 0.3 [-0.06, 0.8] | -0.4 [-5.0, 2.4] | -1.0 [-3.4, 1.4] |
| | Western Europe | -0.06 [-2.3, 1.6] | -0.1 [-0.8, 0.6] | -0.4 [-3.8, 3.4] | -0.4 [-2.2, 1.7] |
| TOTAL_LINEAR_high | Eastern China | 2.0 [-2.9, 7.4] | 0.3 [-0.4, 0.8] | -1.5 [-5.5, 3.9] | -1.7 [-6.7, 1.1] |
| | Eastern U.S. | 1.8 [-1.3, 3.9] | 0.4 [-0.04, 0.6] | -0.4 [-2.7, 1.5] | -1.4 [-5.7, 0.8] |
| | Western Europe | 0.3 [-2.1, 3.3] | 0.03 [-0.6, 1.0] | -0.9 [-4.0, 1.9] | -0.9 [-3.3, 0.9] |
| TOTAL_LINEAR_low | Eastern China | -0.3 [-3.2, 2.6] | 0.03 [-0.5, 0.4] | 0.1 [-3.0, 4.8] | -2.0 [-8.2, 0.4] |
| | Eastern U.S. | 1.1 [-2.5, 3.2] | 0.2 [-0.2, 0.6] | -0.3 [-3.0, 1.6] | -2.0 [-7.4, 0.07] |
| | Western Europe | -0.7 [-3.8, 1.3] | -0.2 [-0.7, 0.2] | -0.3 [-1.8, 2.1] | -0.8 [-4.2, 1.3] |

*2) I further noticed that the Figure captions are often rather short and difficult to understand. Below I give some suggestions for improvements.*

*Figure 1: Please mention what is meant with 'summer' concentrations. I assume June-July-August, 24 hours average.*

**Response:**

The figure caption has been clarified as follows: "Evaluations of simulated summer (June-August) daily (24-h average) surface O3 concentrations in the CTRL run."

*In Figure 2 I do not see any bounding boxes in Africa. Are the parameters 24 hour averaged?*

**Response:**

The bounding boxes in Africa are used to answer the questions raised by Reviewer 1. Since Africa is not the main region that we focus on, we removed the boxes around Africa in the main text.

For the figure caption, we use "summer daily" which has been explained in Figure 1 caption.

*Table S1: explain the meaning of the parameters in Table description.*

**Response:**

We added Table title and related descriptions as follows:

**Table S1** Biophysical parameters for different plant functional types (PFTs) used in YIBs model. PFTs are tundra (TDA), C3 grassland (GRAC3), shrubland (SHR), savanna (SAV), deciduous broadleaf forest (DBF), evergreen needleleaf forest (ENF), evergreen broadleaf forest (EBF), C3 cropland (CROC3) and C4 grassland/cropland (GRAC4/CROC4). *m* and *b* are the slopes and intercepts when calculating stomatal conductance (Eq. (2)). Parameter *a* represents the $O_3$ damaging sensitivity dependent on vegetation types with a range from low (*a_low*) to high (*a_high*) values. $F_{O3,crit}$ is a critical threshold for damage.

*Figure S2: I assume this is the regression of the median of the six member model ensemble. Please mention in the Figure caption.*

**Response:**

This figure shows evaluations of surface $O_3$ concentrations simulated by six models. We did not show the ensemble median of these models. In the figure caption, we clarified as follows:

**Figure S2.** Evaluations of summer (June-August) daily (24-h average) surface $O_3$ concentrations ([$O_3$]) simulated by six models from the ACCMIP dataset. Observed ground-level [$O_3$] are collected from networks of AQMN-MEE in China, CASTNET in U.S. and EMEP in Europe. Simulated surface [$O_3$] are interpolated to the observational sites by using the bilinear interpolation method. The purple line shows the linear regression between observations and simulations for individual models. The black dashed line shows the 1:1 lines.

*Figure S3: spell out IPE in caption, and F-scheme.*

**Response:**

Figure S3 does not include results from F-scheme. The figure caption was corrected as follows:

**Figure S3.** Monthly mean isoprene emissions (IPE) in the CTRL simulation averaged over eastern China, eastern U.S. and western Europe, respectively.

*Figure S5. Explain in caption what is done with the information on boxes. Significance is based on what?*

**Response:**

We clarified as follows:

**Figure S5:** Effects of $O_3$ damage to photosynthesis and stomatal conductance on summertime $O_3$ dry deposition velocity (cm s$^{-1}$) with (a) high and (b) low damaging sensitivities. Dotted grids indicate significant changes ($p<0.05$) due to $O_3$ damages to photosynthesis and stomatal conductance. Intense changes in Eastern China, eastern U.S., and western Europe are highlighted by three green boxes.

*Figure S8. Not clear what is meant with meteorological feedback to IPE. Some additional info (e.g. feedback due to higher temperatures) would help the reader.*

**Response:**

Sorry for the inaccurate description 'meteorological feedback'. The caption has been revised as 'Same as Fig. S5 but for feedbacks on IPE, which are led by the effect of $O_3$ damage to photosynthesis and stomatal conductance.'

We clarified as follows:

**Figure S8.** Same as Fig. S5 but for feedbacks on IPE, which are led by the effect of $O_3$ damage to photosynthesis and stomatal conductance.

*Main text:*
*p. 8 l. 17; reasonably reproduces spatial patterns.*

**Response:**

Corrected as suggested.

*p. 10 l. 28 this sentence is confusing (negative in conjunction with the plus sign). I assume you mean negative changes of 0.9 and 0.7 due to reductions of isoprene emissions under high ozone.*

**Response:**

Sorry for the confusion. The sentence has been clarified as follows:

[revised manuscript text omitted]